# Smooth, exact rotational symmetrization for deep learning on point clouds

**Sergey N. Pozdnyakov and Michele Ceriotti**
Laboratory of Computational Science and Modelling,
Institute of Materials, Ecole Polytechnique Fédérale de Lausanne,
Lausanne 1015, Switzerland
sergey.pozdnyakov@epfl.ch, michele.ceriotti@epfl.ch

## Abstract

Point clouds are versatile representations of 3D objects and have found widespread application in science and engineering. Many successful deep-learning models have been proposed that use them as input. The domain of chemical and materials modeling is especially challenging because exact compliance with physical constraints is highly desirable for a model to be usable in practice. These constraints include smoothness and invariance with respect to translations, rotations, and permutations of identical atoms. If these requirements are not rigorously fulfilled, atomistic simulations might lead to absurd outcomes even if the model has excellent accuracy. Consequently, dedicated architectures, which achieve invariance by restricting their design space, have been developed. General-purpose point-cloud models are more varied but often disregard rotational symmetry. We propose a general symmetrization method that adds rotational equivariance to any given model while preserving all the other requirements. Our approach simplifies the development of better atomic-scale machine-learning schemes by relaxing the constraints on the design space and making it possible to incorporate ideas that proved effective in other domains. We demonstrate this idea by introducing the Point Edge Transformer (PET) architecture, which is not intrinsically equivariant but achieves state-of-the-art performance on several benchmark datasets of molecules and solids. A-posteriori application of our general protocol makes PET exactly equivariant, with minimal changes to its accuracy.

## 1 Introduction

Contrary to 2D images that are well-described by regular and dense pixel grids, 3D objects usually have non-uniform resolution and are represented more naturally by an irregular arrangement of points in 3D. The resulting *point clouds* are widely used in many domains, including autonomous driving, augmented reality, and robotics, as well as in chemistry and materials modeling, and are the input of a variety of dedicated deep-learning techniques[1]. Whenever they are used for applications to the physical sciences, it is desirable to ensure that the model is consistent with fundamental physical constraints: invariance to translations, rotations, and permutation of identical particles, as well as smoothness with respect to geometric deformations[2–4]. In the case of atomistic modeling, the application domain we focus on here, *exact* compliance with these requirements is highly sought after. Lack of smoothness or symmetry breaking are common problems of conventional atomistic modeling techniques[5, 6]. These have been shown in the past to lead to artifacts ranging from numerical instabilities[7] (which are particularly critical in the context of high-throughput calculations[8, 9]) to qualitatively incorrect and even absurd[10, 11] simulation outcomes.

37th Conference on Neural Information Processing Systems (NeurIPS 2023).

These concerns have led to the development of dedicated models for atomistic simulations that rigorously incorporate all these constraints [12–15] by using only symmetry preserving operations. These restrictions limit the design space of these models, for example, leading to the lack of universal-approximation property in many popular methods, as we discuss in Section 2. Furthermore, symmetry requirements prevented the application of models developed in other domains to atomistic simulations. As an illustrative example, one can mention PointNet++[16], an iconic architecture for generic point clouds. This model is not rotationally invariant. Therefore, even though it might have excellent accuracy, it has never been applied to atomistic simulations to the best of our knowledge. It is essential to distinguish between exact, rigorous equivariance and an approximate one, which any model can learn by rotational augmentations. Due to the subtle nature of possible artifacts, an approximate equivariance is considered insufficient.

Out of the mentioned symmetry constraints, the only challenging one is rotational equivariance. As we discuss in Section 4, all the other requirements are either already fulfilled in most functional forms used by existing models for generic point clouds or can be enforced with trivial modifications.

In this paper, we introduce a general symmetrization protocol that enforces exact rotational invariance *a posteriori*, for an arbitrary backbone architecture, without affecting its behavior with respect to all the other requirements. It eliminates the wall between communities, making most of the developments for generic point clouds applicable to atomistic simulations. Furthermore, it simplifies the development of new, more efficient architectures by removing the burden of incorporating the exact rotational invariance into their functional form. As an illustration, we design a model named Point Edge Transformer (PET) that achieves state-of-the-art performance on several benchmarks ranging from high-energy $CH_4$ configurations to a diverse collection of small organic molecules to the challenging case of periodic crystals with dozens of different atomic species. Being not intrinsically rotationally equivariant, PET has benefited from the enlarged design space. Our symmetrization method a posteriori makes it rigorously equivariant and, thus, applicable to atomistic simulations.

## 2  Equivariant models and atomic-scale applications

Models used in chemical and materials modeling achieve rotational invariance by two main mechanisms. The first involves using only invariant internal coordinates such as interatomic distances, angles, or even dihedral angles[17]. The second involves using equivariant hidden representations that transform in a predictable manner under symmetry operations. For example, some intermediate activations in a neural network can be expressed as vectors that rotate together with the input point cloud[18]. This approach restricts the design space to only such functional forms that preserves the equivariance.

**Local decomposition.**  Models in atomistic machine learning often rely on the prediction of local properties associated with atom-centered environments $A_i$, either because they are physical observables (e.g., NMR chemical shieldings[19, 20]) or because they provide a decomposition of a global extensive property, such as the energy, into a sum of atomic contributions, $y = \sum_i y(A_i)$. Here, $y(A_i)$ indicates a learnable function of the environment of the $i-$th atom, defined as all the neighbors within the cutoff radius $R_c$. This local decomposition is rooted in physical considerations[21] and is usually beneficial for the transferability of the model.

**Local invariant descriptors.**  The classical approach to construct approximants for $y(A_i)$ is to first compute smooth, invariant descriptors for each environment[14, 22–27] and then feed them to a conventional machine learning model. Such models can be 1) linear regression[27], 2) kernel regression[28], or 3) feed-forward neural network with a smooth activation function.[29].

**Distance-based message-passing.**  Another popular method involves constructing a molecular graph and feeding it to a Graph Neural Network (GNN). Early, and very popular, models rely on invariant two-body messages based only on interatomic distances[30, 31]. In this case, a molecular graph is constructed by representing all the atoms by nodes, drawing edges between all the atoms within a certain cutoff radius, and decorating edges with the Euclidean distance between the corresponding atoms.

Recently, it was discovered that such models are not universal approximators. Specifically, they cannot distinguish between certain atomic configurations[32]. This issue can be directly attributed to the restricted design space of rotationally equivariant models. In an enlarged design space, one

could decorate the edges of a molecular graph with x, y, and z components of the corresponding displacement vectors. This would immediately ensure the universal approximality of the model. However, the necessity to enforce rotational equivariance of the predictions dictates decorating the edges of a molecular graph only with rotationally invariant Euclidean distances. Such a restriction severely limits the expressiveness of the corresponding models.

**Higher-order message-passing.** Subsequently, more expressive models were developed. For example, some of them use angles[33], or even dihedrals[17], in addition to distances between the atoms. Others, such as Tensor Field Network[34], Nequip[35], MACE[36], SE(3) transformers[37], employ SO(3) algebra to maintain equivariance of hidden representations. These models solve the incompleteness issue of simple atom-centered geometric descriptors and distance-based GNNs. The quality of a model, however, doesn't reduce to the simple presence of a universal approximation behavior. An extended design space can still be beneficial to obtain more accurate or efficient models.

## 3 Models for generic point clouds

Point clouds have found many applications beyond atomistic modeling. For example, detectors such as LiDARs represent the scans of the surrounding world as collections of points. Multiple methods have been developed for such domains. Contrary to atomistic machine learning, there are no such strict symmetry requirements for practical applications. Consequently, most models do not exactly incorporate rotational equivariance and rely instead on rotational augmentations.

Many successful models based on 2D projections of the cloud have been proposed in computer vision [38–41]. Here, we focus on explictly 3D models, that are most relevant for chemical applications.

**Voxel-based methods.** The complete 3D geometric information for a structure can be encoded in a permutation-invariant manner by projecting the point cloud onto a regular 3D voxel grid, and further manipulated by applying three-dimensional convolutions. The computational cost of a naive approach, however, scales cubically with the resolution, leading to the development of several schemes to exploit the sparsity of the voxel population[42–44], which is especially pronounced for high resolutions.

**Point-based methods.** By extending the convolution operator to an irregular grid, one can avoid the definition of a fixed grid of voxels. For instance[45, 46], one can evaluate an expression such as

$$(\mathcal{F} * g)(\mathbf{r})_m = \sum_i \sum_n^{N_{in}} g_{mn}(\mathbf{r}_i - \mathbf{r}) f_n^i, \tag{1}$$

where, $f_n^i$ is the $n$-th feature of the point $i$, and $g$ is a collection of $N_{in}N_{out}$ learnable three-dimensional functions, where $N_{in}$ and $N_{out}$ are the numbers of input and output features respectively. The output features $(\mathcal{F} * g)(\mathbf{r})_m$ can be evaluated at an arbitrary point $\mathbf{r}$, allowing for complete freedom in the construction of the grid. PointNet[47] is a paradigmatic example of an architecture that eliminates the distinction between point features and positions. The core idea of these approaches is that an expression such as $f(\mathbf{r}_1, \mathbf{r}_2, ..., \mathbf{r}_n) = \gamma(\max_i(\{h(\mathbf{r}_i)\}))$, where $\gamma$ and $h$ are learnable functions, can approximate to arbitrary precision any continuous function of the point set $\{\mathbf{r}_i\}$. PointNet++[16] and many other models[46, 48, 49] apply similar functional form in a hierarchical manner to extract features of local neighborhoods. Many of these methods have graph-convolution or message-passing forms, similar to those discussed in Section 2, even though they do not enforce invariance or equivariance of the representation. Many transformer models for point clouds have also been proposed[50–53], that incorporate attention mechanisms at different points of their architecture.

## 4 Everything but rotational equivariance

The generic point-cloud models discussed in the previous section do not incorporate all of the requirements (permutation, translation and rotation symmetry as well as smoothness) that are required by applications to atomistic simulations. Most models, however, do incorporate everything but rotational invariance, or can be made to with relatively small modifications. For instance, translational invariance can be enforced by defining the grids or the position of the points in a coordinate system that is relative to a reference point that is rigidly attached to the object.

Another common problem of many of these methods is the lack of smoothness. However, most architectures can be made differentiable with relative ease, with the exception of very few operations

such as downsampling via farthest point sampling[16]. For example, derivative discontinuities associated with non-smooth activation functions, such as ReLU[54], can be eliminated simply by using a smooth activation functions instead[55–58]. A slightly less trivial problem arises for models using the convolutional operator of Eq. (1), that introduces discontinuities related to (dis)appearance of new points at the cutoff sphere, even for smooth functions $g_{mn}$. These discontinuities can be removed by modifying the convolutional operator as:

$$(\mathcal{F} * g)(\mathbf{r})_m = \sum_i^{N_{in}} \sum_n g_{mn}(\mathbf{r}_i - \mathbf{r}) f_n^i f_c(\|\mathbf{r}_i - \mathbf{r}\| | R_c, \Delta_{R_c}), \tag{2}$$

where the cutoff function $f_c$, illustrated in Fig. 5c, can be chosen as an infinitely differentiable switching function that smoothly zeroes out contributions from the points approaching the cutoff sphere, and the parameter $\Delta_{R_c} > 0$ controls how fast it converges to 1 for $r < R_c$. A similar strategy can be applied to PointNet-like methods. 3D CNNs on regular grids that use smooth activation and pooling layers such as sum or average operations are also smooth functions of the voxel features. Including also a smooth projection of the point cloud on the voxel grid suffices to make the overall methods smooth. We further discuss in the Appendix G examples of such smooth projections, along with other modifications that can be applied to make the general point-cloud architectures discussed in Sec. 3 differentiable. Finally, most – but not all[59] – of the generic models for point clouds are invariant to permutations.

In summary, there is a wealth of point-cloud architectures that have proven to be very successful in geometric regression tasks, and are, or can be easily made, smooth, permutation and translation invariant. The main obstacle that hinders their application to tasks that require a fully-symmetric behavior, such as those that are common in atomistic machine learning, is invariance or covariance under rigid rotations. The ECSE protocol presented in the next section eliminates this barrier.

## 5 Equivariant Coordinate System Ensemble

Our construction accepts a smooth, permutationally, translationally, but not necessarily rotationally invariant backbone architecture along with a point that is rigidly attached to the point cloud, and produces an architecture satisfying all four requirements. While in principle, there are many choices of the reference point, we will focus for simplicity on the case in which it is one of the nodes in the point cloud. More specifically, we formulate our method in the context of a local decomposition of the (possibly tensorial) target $\mathbf{y}(A) = \sum_i \mathbf{y}(A_i)$ where the reference point is given by the position of the central atom $\mathbf{r}_i$, and the model estimates an atomic contribution $\mathbf{y}(A_i)$ given the local neighborhood. We name our approach Equivariant Coordinate System Ensemble (ECSE, pron. eʃe).

**Local coordinate systems.** A simple idea to obtain a rotationally equivariant model would be to define a coordinate system rigidly attached to an atomic environment. Next, one can express Cartesian coordinates of all the displacement vectors from the central atom to all the neighbors and feed them to any, possibly not rotationally invariant, backbone architecture. Since the reference axes rotate together with the atomic environment, the corresponding projections of the displacement vectors are invariant with respect to rotations, and so is the final prediction of the model.

This approach can be applied easily to rigid molecules[60, 61], but in the general case it is very difficult to define a coordinate system in a way that preserves smoothness to atomic deformations. For example, the earliest version of the DeepMD framework[62] used a coordinate system defined by the central atom and the two closest neighbors. Smooth distortions of the environment can change the selected neighbors, leading to a discontinuous change of the coordinate systems and therefore to discontinuities in the predictions of the model. For this reason, later versions of DeepMD[63] switched to descriptors that can be seen as a close relative of Behler-Parrinello symmetry functions[22], which guarantee smoothness and invariance.

**Ensemble of coordinate systems.** The basic formulation of our symmetrization protocol, illustrated in Fig. 5a, is simple: using *all* the possible coordinate systems defined by all pairs of neighbors instead of singling one out, and averaging the predictions of a non-equivariant model over this ensemble of reference frames:

$$\mathbf{y}_S(A_i) = \sum_{jj' \in A_i} w_{jj'} \hat{R}_{jj'} [\mathbf{y}_0(\hat{R}_{jj'}^{-1}[A_i])] \Big/ \sum_{jj' \in A_i} w_{jj'}, \tag{3}$$

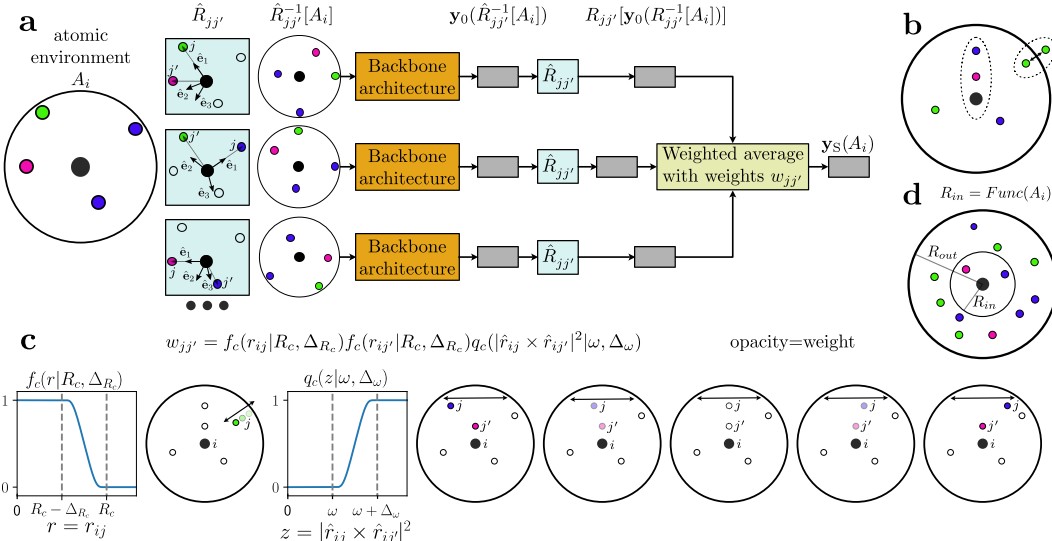

Figure 1: (a) Equivariant coordinate-system ensemble: Each ordered pair of neighbors defines a local coordinate system. Next, an atomic environment is projected on all of them (which is equivalent to rotation) and used as input for a backbone architecture. If outputs are covariant, such as vectors, they are rotated back to the initial coordinate system. Finally, predictions are averaged over. (b) Discontinuities related to plain average. The weighted average with weights $w_{jj'}$ resolves these problems. (c) Cutoff functions $f_c$ and $q_c$ used to define weights $w_{jj'}$ (d) To reduce the computational cost, an adaptive cutoff $R_{in}$ is used, which adjusts to a given geometry instead of being a global user-specified constant.

where $\mathbf{y}_S$ indicates the symmetrized model, $\mathbf{y}_0$ the initial (non-equivariant) backbone architecture, and $\hat{R}_{jj'}[\cdot]$ indicates the rotation operator for the coordinate system defined by the neighbors $j$ and $j'$ within the $i$-centered environment $A_i$. In other words, $\hat{R}_{jj'}^{-1}[A_i]$ is an atomic environment expressed in the coordinate system defined by neighbors $j$ and $j'$. The summation is performed over all the ordered pairs of neighbors within a cutoff sphere with some cutoff radius $R_c$.

Given a pair of neighbors with displacement vectors from central atom $\mathbf{r}_{ij}$ and $\mathbf{r}_{ij'}$ the corresponding coordinate system consists of the vectors $\hat{r}_{ij}$, $\hat{r}_{ij} \times \hat{r}_{ij'}$, and $\hat{r}_{ij} \times [\hat{r}_{ij} \times \hat{r}_{ij'}]$. If the model predicts a vectorial (or tensorial) output, it is rotated back into the original coordinate system by an outer application of operator $\hat{R}_{jj'}$. Thus, our symmetrization scheme allows for getting not only invariant predictions, but also covariant ones, such as vectorial dipole moments. It is worth to note that the idea of using all the possible coordinate systems has also been used to define a polynomially-computable invariant metric to measure the similarity between crystal structres[64–66], and to construct provably complete invariant density-correlation descriptors [67].

Fig. 5b depicts the necessity to use the weighted average with weights $w_{jj'}$ instead of a plain one. The use of plain average in Eq. 3 would lead to the lack of smoothness. Indeed, if a new atom enters the cutoff sphere, it immediately yields new terms into the summation in eq. 3, which would lead to a discontinuous gap in predictions. Furthermore, if two neighbors and a central atom appear to be collinear, the associated coordinate system is ill-defined.

Both issues can be solved by introducing a mechanism in which each coordinate system is assigned a different weight, depending on the positions $(\mathbf{r}_{ij}, \mathbf{r}_{ij'})$ of the neighbors that define the reference frame:

$$w_{jj'} = w(\mathbf{r}_{ij}, \mathbf{r}_{ij'}) = f_c(r_{ij}|R_c, \Delta_{R_c}) f_c(r_{ij'}|R_c, \Delta_{R_c}) q_c(|\hat{r}_{ij} \times \hat{r}_{ij'}|^2|\omega, \Delta_\omega), \qquad (4)$$

where the cutoff functons $f_c$ and $q_c$ are illustrated in Fig. 5c. Thanks to the presence of $f_c$, the terms in Eq. (3) that are associated with atoms entering or exiting the cutoff sphere have zero weights. Similarly, $q_c$ ensures that pairs of neighbors that are nearly collinear do not contribute to the symmetrized prediction. One last potential issue is the behavior when *all* pairs of neighbors are

(nearly) collinear, which would make Eq. (3) ill-defined, falling to $\frac{0}{0}$ ambiguity. We discuss in the Appendix F.3 two possible solutions for this corner case.

**Adaptive cutoff.**  To make the ECSE protocol practically feasible, it is necessary to limit the number of evaluations of the non-equivariant model, given that the naive number of evaluations grows quadratically with the number of neighbors. An obvious consideration is that there is no reason why the cutoff radius used by ECSE should be the same as the one used by the backbone architecture. Thus, one can achieve significant computational savings by simply defining a smaller cutoff for symmetrization. However, particularly for inhomogeneous point cloud distributions, a large cutoff may still be needed to ensure that all environments have at least one well-defined coordinate system. For this reason, we introduce an adaptive inner cutoff $R_{\text{in}}(A_i)$, which is determined separately for each atomic environment $A_i$ instead of being a global, user-specified constant. Eq. (3) requires that at least one pair of neighbors is inside the cutoff sphere. Simultaneously, it is desirable to make it as small as possible for computational efficiency. Thus, it makes sense to define $R_{\text{in}}(A_i)$ along the lines of the distance from the central atom to the second closest neighbor. It should be larger, but not much larger, than the second-nearest-neighbor distance.

Our definition of $R_{\text{in}}(A_i)$, given in the Appendix F.4, is inspired by this simple consideration, and has the following properties: (1) Contrary to the naive second-neighbor distance, $R_{\text{in}}(A_i)$ depends smoothly on the positions of all the atoms. (2) It encompasses at least one non-collinear pair of neighbors, which yields at least one well-defined coordinate system: the functional form is chosen so that if $k$ nearest neighbors are collinear, $R_{\text{in}}(A_i)$ is expanded to enclose the $k + 1$-th for any $k$.

With such an adaptive cutoff, only a few pairs of neighbors are used to construct a coordinate system. Thus, this construction greatly reduce the cost of evaluating the ECSE-symmetrized model.

**Training and symmetrization.**  The most straightforward way to train a model using the ECSE protocol is to apply it from the very beginning and train a model that is overall rotationally equivariant. This approach, however, increases the computational cost of training, since applying ECSE entails multiple evaluations of the backbone architecture. An alternative approach, which we follow in this work, is to train the backbone architecture with rotational augmentations, and apply ECSE only for inference. Given that ECSE evaluates the weighted average of the predictions of the backbone architecture, this increases the cost of inference, but may also increase the accuracy of the backbone architecture, playing the role of test augmentation.

**Message-passing.**  Message-passing schemes can be regarded as local models with a cutoff radius given by the receptive field of the GNN, and therefore ECSE can be applied transparently to them. In a naïve implementation, however, the same message would have to be computed for the coordinate systems of all atoms within the receptive field. This implies a very large overhead, even though the asymptotic cost remains linear with system size. An alternative approach involves symmetrizing the outgoing messages with the ECSE protocol. This requires a deeper integration within the model, that also changes the nature of the message-passing step, given that one has to specify a transformation rule to be applied to messages computed in different coordinate systems. We discuss in the Appendix F.10 the implications for the training strategy, and some possible workarounds.

**Related work.**  There are several approaches, such as the earliest version of the DeepMD framework[62], discussed above, that use local coordinate systems to enforce rotational equivariance for any backbone architecture. The frame averaging (FA) framework [68, 69] proposes to use the eigenvectors of the centered covariance matrix to define local coordinate systems, which leads to discontinuities every time the eigenvalues coincide with each other. To the best of our knowledge, ECSE is the first general symmetrization method allowing to enforce rotational equivariance while preserving smoothness, which is a highly desirable property for atomistic simulations.

Finally, a completely different approach is introduced by the vector neurons framework[70] that converts arbitrary backbone architecture into an equivariant one by enforcing all hidden representations in the model to transform as vectors with respect to rotations.

# 6   Point Edge Transformer

An obvious application of the ECSE protocol would be to enforce equivariance on some of the point-cloud architectures discussed in Section 3, making them directly-applicable to atomistic modeling and to any other domain with strict symmetry requirements. Here we want instead to focus on a less-

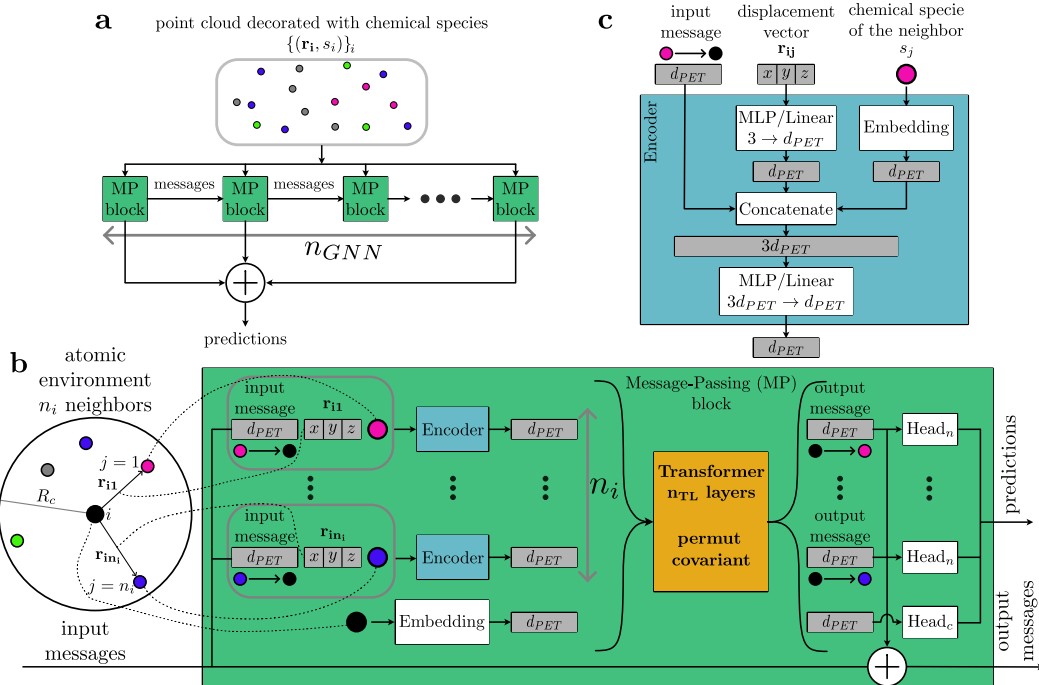

Figure 2: Architecture of the Point-Edge Transformer (PET). White and colored boxes represent layers; gray boxes and lines represent data. (a) PET is a message-passing architecture. At each of the $n_{\text{GNN}}$ message-passing (MP) interactions, messages are communicated between all the pairs of atoms closer than a certain cutoff distance $R_c$. At each stage, the corresponding MP block computes output messages and predictions of the target property given the input messages and geometry of the point cloud. (b) For each atom in the system, we define atom-centered environment $A_i$ as a collection of all the neighbors within the cutoff distance $R_c$. The MP block is applied to each such atomic environment. Given 1) the geometry of the atomic environment, 2) the chemical species of the atoms, and 3) input messages from all the neighbors to the central atom it produces output messages from the central atom to all the neighbors and contribution to the prediction of the target property. The first step is to encode all the information associated with each neighbor to an abstract token of dimensionality $d_{\text{PET}}$. Next, the collection of such tokens (with the one associated with the central atom) is fed into the transformer with $n_{\text{TL}}$ self-attention layers. The transformer does permutationally covariant transformation. Thus, the association between the tokens and neighbors is preserved. Therefore, we can simply treat output tokens as output messages to the corresponding neighbors. (c) The Encoder layer first maps all the sources of information into dimensionality $d_{\text{PET}}$. Next, all 3 tokens are concatenated and compressed into a single one of the desired size.

obvious, but perhaps more significant, application: demonstrating that lifting the design constraint of rotational equivariance makes it possible to construct better models. To this end, we introduce a new architecture that we name Point Edge Transformer (PET), which is built around a transformer that processes edge features and achieves state-of-the-art accuracy on several datasets that cover multiple subdomains of atomistic ML. Due to the focus on edge features, PET shares some superficial similarities with the Edge Transformer[71] developed for natural language processing, Allegro (a strictly local invariant interatomic potential[72] that loosely resembles a single message-passing block of PET) as well as with early attempts by Behler et al. applying MLPs to invariant edge features[73].

The PET architecture is illustrated in Fig. 2. More details are given in Appendix A. The core component of each message-passing block is a permutationally-equivariant transformer that takes a set of tokens of size $d_{\text{PET}}$ associated with the central atom and all its neighbors, and generates new tokens that are used both to make predictions and as output messages. The transformer we use in PET is a straightforward implementation of the classical one[74], with a modification to the attention mechanism that ensures smoothness with respect to (dis)appearance of the neighbors at the

cutoff radius. The attention coefficients $\alpha_{ij}$ determining the contribution from token $j$ to token $i$ are modified as $\alpha_{ij} \leftarrow \alpha_{ij} f_c(r_j | R_c, \Delta_{R_c})$, and then renormalized.

One of the key features of PET is that it operates with features associated with each *edge*, at variance with other deep learning architectures for point clouds that mostly operate with vertex features. Whereas the calculation of new vertex features by typical GNNs involves aggregation over the neighbors, the use of edge features allows for the construction of an *aggregation-free* message-passing scheme, avoiding the risk of introducing an information bottleneck.

The transformer itself can be treated as a GNN[75], thus our architecture can be seen as performing *localized message passing*, increasing the complexity of local interactions it can describe, while avoiding an over-increase of the receptive field. The latter is undesirable because it makes parallel computation less efficient, and thus hinders the application of such models to large-scale molecular dynamics simulations[72].

Since transformers are known to be universal approximators[76], so is *each* of the message-passing blocks used in PET.

**Limitations.** As discussed in Section 5, the most efficient application of the ECSE protocol on top of a GNN requires re-designing the message-passing mechanism. For the current moment, our proof-of-principle implementation follows a simpler approach (see more details in Appendix F.11) and favors ease of implementation instead of computational efficiency (e.g., all weights and most of the intermediate values used in the ECSE scheme are stored in separate zero-dimensional PyTorch[77] tensors). This leads to a significant (about 3 orders of magnitude) overhead over the inference of the backbone architecture. The base PET model, however, gains efficiency by operating directly on the Cartesian coordinates of neighbors. Even with this inefficient implementation of ECSE, exactly equivariant PET inference is much cheaper than the reference first-principles calculations.

## 7    Benchmarks

We benchmark PET and the ECSE scheme over six different datasets, which have been previously used in the literature and which allow us to showcase the performance of our framework, and the ease with which it can be adapted to different use cases. The main results, are compared with the state of the art in Figure 3 and Table 1, while in-depth analyses can be found in the Appendix C.

As a first benchmark, we conduct several experiments with the liquid-water configurations from Ref. 85. This dataset is representative of those used in the construction of interatomic potentials, and presents interesting challenges in that it contains distorted structures from path integral molecular dynamics and involves long-range contributions from dipolar electrostatic interactions. One of the key features of PET is the possibility to increase the expressive power by either adding MP blocks or by making transformers in each block deeper. Panel (a) in Fig. 3 shows that increasing the number of GNN blocks improves the accuracy of PET, and that for a given number of blocks, a shallow transformer with a single layer performs considerably worse than one with two or more layers. Given that stacking multiple GNN blocks increases both the receptive field and the flexibility in describing local interactions, it is interesting to look at the trend for a fixed total number of transformer layers (Fig. 3b), that shows that a $6 \times 2$ model outperforms a large $12 \times 1$ stack of shallow transformers, even though the latter has additional flexibility because of the larger number of heads and pre-processing units. With the exception of the shallower models, PET reduces the error over the state-of-the-art equivariant model NEQUIP[35] by $\sim 30\%$. In problems that are less dependent on long-range physics, it may be beneficial to increase the depth of transformers and reduce the number of MP blocks. The accuracy of the model can also be further improved by extending $R_c$ beyond 3.7Å (a value we chose to ensure approximately 20 neighbors on average), reaching a force MAE of 14.4 meV/Å when using a cutoff of 4.25Å.

We then move to the realm of small molecules with the COLL dataset[86], that contains distorted configurations of molecules undergoing a collision. COLL has been used extensively as a benchmark for chemical ML models, in particular for GNNs that employ information on angles and dihedrals to concile rotaional invariance and universal approximation[17, 87]. PET improves by $\sim 13\%$ the error on forces (the main optimization target in previous studies) while reducing by a factor of 4 the error on atomization energies relative to the respective state of the art. In order to assess the accuracy of PET for extreme distortions, and a very wide energy range, we consider the database of $CH_4$

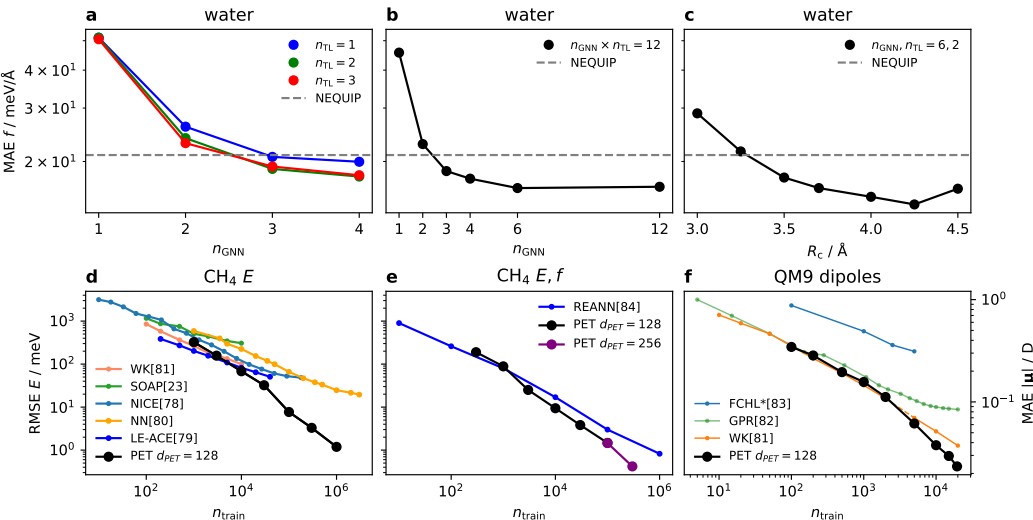

Figure 3: (a-c) Accuracy of PET potentials ($y_0$) of liquid water, compared with NEQUIP[35]. (a) Accuracy for different numbers of message-passing blocks $n_{GNN}$ and transformer layers $n_{TL}$; (b) Accuracy as a function of $n_{GNN}$, for constant $n_{GNN} \times n_{TL} = 12$.; (c) Accuracy as a function of cutoff. (d-f) Learning curves for different molecular data sets, comparing symmetrized PET models ($y_S$) with several previous works[23, 78–84], including the current state of the art. (d) Random $CH_4$ dataset, training only on energies; (e) Random $CH_4$ dataset, training on energies and forces; (f) Vectorial dipole moments in the QM9 dataset[82].

configurations first introduced in Ref. 80. This dataset contain random arrangements of one C and 4 H atoms, that are only filtered to eliminate close contacts. The presence of structures close to those defying C-centered angle-distances descriptors adds to the challenge of this dataset. The learning curves for PET in Figure 3c and d demonstrate the steep, monotonic improvement of performance for increasing train set size, surpassing after a few thousand training points the best existing models (a linear many-body potential for the energy-only training[81] and REANN, an angle-based GNN[84] for combined energy/gradient training).

Table 1: Comparison of the accuracy of PET and current state-of-the-art models for the COLL, MnO, HM21 and HEA data sets. A more comprehensive comparison with leading models is provided in the Appendix C. Energy errors are given in meV/atom, force errors in meV/Å. In all cases the PET model is nearly equivariant, and the difference between the accuracy of $y_0$ and $y_S$ is minuscule. For HME21, we report error bars over 5 random seeds, and results for an ensemble of symmetrized models.

| dataset | COLL | | MnO | | HME21 | | HEA | |
|---|---|---|---|---|---|---|---|---|
| metric | MAE $f$ | MAE $E$ | RMSE $f$ | RMSE $E$/at. | MAE $|f|$ | MAE $E$/at. | MAE $f$ | MAE $E$/at. |
| SOTA | 26.4[17] | 47[86] | 125[88] | 1.11[88] | 138[89] | 15.7[89] | 190[90] | 10[90] |
| model | GemNet | DimeNet++ | mHDNNP | | MACE | | HEA25-4-NN | |
| PET ($y_0$) | 23.1 | 12.0 | 22.7 | 0.312 | $140.5 \pm 2.0$ | $17.8 \pm 0.1$ | 60.2 | 1.87 |
| PET ($y_S$) | 23.1 | 11.9 | 22.7 | 0.304 | $141.6 \pm 1.9$ | $17.8 \pm 0.1$ | 60.1 | 1.87 |
| PET (ens.) | | | | | 128.5 | 16.8 | | |

The PET performs well even for condensed-phase datasets that contain dozens of different atomic types. The HEA dataset contains distorted crystalline structures with up to 25 transition metals *simultaneously*[90]. Ref. 90 performs an explicit compression of chemical space, leading to a model that is both interpretable and very stable. The tokenization of the atomic species in the encoder layers of PET can also be seen as a form of compression. However, the more flexible form of the model compared to HEA25-4-NN allows for a a 3(5)–fold reduction of force(energy) hold-out errors. The model, however, loses somewhat in transferability: in the extrapolative test on high-temperature MD trajectories suggested in Ref. 90, PET performs less well than HEA25-4-NN (152 vs 48 meV/at.

MAE at 5000 K) even though room-temperature MD trajectories are better described by PET. The case of the HME21 dataset, which contains high-temperature molecular-dynamics configurations for structures with a diverse composition, is also very interesting. PET outperforms most of the existing equivariant models, except for MACE[89], which incorporates a carefully designed set of physical priors[36], inheriting much of the robustness of shallow models. PET, however, comes very close: the simple regularizing effect of a 5-models PET ensemble is sufficient to tip the balance, bringing the force error to 128.5 meV/at. Another case in which PET performs well, but not as well as the state of the art, is for the prediction of the atomization energy of molecules in the QM9 dataset[91]. PET achieves a MAE of 6.7 meV/molecule, in line with the accuracy of DimeNet++[87], but not as good as the 4.3 meV/molecule MAE achieved by Wigner kernels[81].

Finally, we consider two cases that allow us to showcase the extension of PET beyond the prediction of the cohesive energy of molecules and solids. The MnO dataset of Eckhoff and Behler[88] includes information on the colinear spin of individual atomic sites, and demonstrates the inclusion of information beyond the chemical nature of the elements in the description of the atomic environments. The improvement with respect to the original model is quite dramatic (Tab. 1). Finally, the QM9 dipole dataset[82] allows us to demonstrate how to extend the ECSE to targets that are covariant, rather than invariant. This is as simple as predicting the Cartesian components of the dipole in each local coordinate system, and then applying to the prediction the inverse of the transformation that is applied to align the local coordinate system (cf. Eq. (3)). The accuracy of the model matches that of a recently-developed many-body kernel regression scheme[81] for small dataset size, and outperforms it by up to 30% at the largest train set size.

# 8   Discussion

In this work, we introduce ECSE, a general method that enforces rotational equivariance for any backbone architecture while preserving smoothness and invariance with respect to translations and permutations. To demonstrate its usage, we also develop the PET model, a deep-learning architecture that is not intrinsically rotationally invariant but achieves state-of-the-art results on several datasets across multiple domains of atomistic machine learning. The application of ECSE makes the model comply with all the physical constraints required for atomistic modeling.

We believe our findings to be important for several reasons. On the one hand, they facilitate the application of existing models from geometric deep learning to domains where exact equivariance is a necessary condition for practical applications. On the other, they challenge the notion that equivariance is a necessary ingredient for an effective ML model of atomistic properties. The state-of-the-art performance of PET on several benchmarks reinforces early indications that non-equivariant models trained with rotational augmentation can outperform rigorously equivariant ones. Similar observations were also made independently by other groups. Spherical Channel Network[92] and ForceNet[93] achieve excellent performance on the Open Catalyst dataset[94] while relaxing the exact equivariance of the model.

It is worthwhile to mention that in the other domains involving point clouds, the application of not invariant models fitted with rotational augmentations is a predominant approach. As an example, one can examine the models proposed for, e.g., the ModelNet40 dataset[95], a popular benchmark for point cloud classification. Even though the target is rotationally invariant, most of the developed architectures are not rigorously equivariant, in the exact sense, as discussed in Section. 2. This provides considerable empirical evidence that the use of not rigorously equivariant architectures with rotational augmentations might be more efficient.

The ECSE method we propose in our work allows to symmetrize *exactly*, and a-posteriori, any point-cloud model, making them suitable for all applications, such as atomistic modeling, which have traditionally relied on symmetry to avoid qualitative artifacts in simulations. This shall facilitate the translation of methodological advances from other domains to atomistic modeling and the implementation into generic point-cloud models of physically-inspired inductive biases (such as smoothness, range, and body order of interactions) that have this far been conflated only with a specific class of equivariant architectures.

## 9 Acknowledgements

We thank Marco Eckhoff and Jörg Behler for sharing the MnO dataset with the collinear spins and Artem Malakhov for useful discussions.

This work was supported by the Swiss Platform for Advanced Scientific Computing (PASC) and the NCCR MARVEL, funded by the Swiss National Science Foundation (SNSF, grant number 182892).

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

# Appendices

## A   Point Edge Transformer

In this section we provide additional details on the specific implementation of the PET model that we use in this work. As discussed in Section D, we provide a reference implementation that can provide further clarification on specific details of the architecture.

The transformer that is the central component of PET is complemented by several pre- and post-processing units. Two embedding layers, $E_c^{(k)}$ and $E_n^{(k)}$, are used to encode the chemical species of the central atom and all the neighbors, respectively. A $3 \rightarrow d_{\text{PET}}$ linear layer $L^{(k)}$ is used to map the 3-dimensional Cartesian coordinates of displacement vectors $\mathbf{r}_{ij}$ to feature vectors of dimensionality $d_{\text{PET}}$, which are further transformed by an activation function $\sigma$ (we use SiLU[96] for all our experiments) to form a position embedding layer $E_r^{(k)} = \sigma(L^{(k)}(\mathbf{r}_{ij}))$. An MLP with one hidden layer, denoted as $M^{(k)}$, with dimensions $3d_{\text{PET}} \rightarrow d_{\text{PET}}$, is used to compress the encoding of the displacement vector, chemical species, and the corresponding input message into a single feature vector containing all this information. The input token fed to the transformer associated with neighbor $j$ is computed as follows:

$$t_j^{(k)} = M^{(k)} \Big[ \text{concatenate}(\tilde{x}_{ij}^{(k-1)}, E_n^{(k)}(s_j), E_r^{(k)}(\mathbf{r}_{ij})) \Big], \tag{5}$$

where $\tilde{x}_{ij}^{(k-1)}$ denotes input message from atom $j$ to the central atom $i$, $s_j$ is the chemical species of neighbor $j$. For the very first message-passing block, there are no input messages, and so $M^{(1)}$ performs a $2d_{\text{PET}} \rightarrow d_{\text{PET}}$ transformation. The token associated with the central atom is given simply as $E_c^{(k)}(s_i)$.

In addition, each message-passing block has two post-processing units: two heads $H_c^{(k)}$ and $H_n^{(k)}$ to compute atomic and bond contributions to the total energy or any other target *at each message passing block*. Atomic contributions are computed as $H_c^{(k)}(x_i^{(k)})$, where $x_i^{(k)}$ is the output token associated with central atom after application of transformer. Bond (or edge) contributions are given as $H_n^{(k)}(x_{ji}^{(k)}) f_c(r_{ij}|R_c, \Delta_{R_c})$, where $x_{ji}^{(k)}$ indicates the output token associated with the $j$-th neighbor. We modulate them with $f_c(r_{ij}|R_c, \Delta_{R_c})$ in order to ensure smoothness with respect to (dis)appearance of atoms at the cutoff sphere. The output tokens are also used to build outgoing messages. We employ the idea of residual connections[97] to update the messages. Namely, the output tokens are summed with the previous messages.

To extend the model to describe atomic properties, such as the collinear atomic spins, we simply concatenate the value associated with neighbor $j$ to the three-dimensional vector with Cartesian components of $\mathbf{r}_{ij}$. Thus, the linear layer $L^{(k)}$ defines a $4 \rightarrow d_{\text{PET}}$ transformation, in contrast to the $3 \rightarrow d_{\text{PET}}$ of a standard model. Additionally, we encode the property associated with the central atom $i$ into the token associated with the central atom. This is done in a manner similar to Eq. (5).

There is great flexibility in the pre-and post-processing steps before and after the application of the transformer. Our implementation supports several modifications of the procedure discussed above, which are controlled by a set of microarchitectural hyperparameters. For example, one can average bond contributions to the target property instead of summing them. In this case, the total contribution from all the bonds attached to central atom $i$ is given as:

$$y_i^{\text{bond }(k)} = \frac{\sum_{j \in A_i} H_n^{(k)}(x_{ji}^{(k)}) f_c(r_{ij}|R_c, \Delta_{R_c})}{\sum_{j \in A_i} f_c(r_{ij}|R_c, \Delta_{R_c})} \tag{6}$$

rather than

$$y_i^{\text{bond }(k)} = \sum_{j \in A_i} H_n^{(k)}(x_{ji}^{(k)}) f_c(r_{ij}|R_c, \Delta_{R_c}). \tag{7}$$

Another possibility is to not explicitly concatenate embeddings of neighbor species in Eq. (5), and instead define the input messages for the very first message-passing block as these neighbor species embeddings. This change ensures that all $M^{(k)}$ have an input dimensionality of $2d_{\text{PET}}$ across all the message-passing blocks. A brief description of these hyper-parameters accompanies the reference PET implementation that we discuss in Section D.

# B  Details of the training protocol

In this section, we provide only the most important details related to benchmarks reported in the main text. The complete set of settings is provided in electronic form, together with the code we used to train and validate models. See Appendix D for details.

**Self-contributions.**  We pre-process all our datasets by subtracting atomic self-contributions from the targets. These atomic terms often constitute a large (if trivial) fraction of the variability of the targets, and removing them stabilizes the fitting procedure. Self-contributions are obtained by fitting a linear model on the *train* dataset using "bag of atoms" features. For example, the COLL dataset contains 3 atomic species - H, C, and O. The "bag of atoms" features for an $H_2O$ molecule are $[2, 0, 1]$. During inference, self-contribution are computed using the weights of the linear regression and are added to the predictions of the main model.

**Loss.**  For a multi-target regression, which arises when fitting simultaneously on energies and forces, we use the following loss function:

$$L = w_E \frac{(\tilde{E} - E)^2}{\mathrm{MSE}_E} + \frac{\frac{1}{3N} \sum_{i\alpha} (-\frac{\partial \tilde{E}}{\partial \mathbf{r}_{i\alpha}} - F_{i\alpha})^2}{\mathrm{MSE}_F}, \tag{8}$$

where $E$ and $F$ are the ground-truth energies and forces, respectively, $\tilde{E}$ is the energy prediction given by the model, $N$ is the number of atoms in the sample, $\alpha$ runs over $x, y, z$, and $\mathrm{MSE}_E$ and $\mathrm{MSE}_F$ are the exponential moving averages of the Mean Squared Errors in energies and forces, respectively, on the validation dataset. We update $\mathrm{MSE}_E$ and $\mathrm{MSE}_F$ once per epoch. We found that for such a loss definition the choice of the dimensionless parameter $w_E$ is crelatively robust, with the best results achieved approximately for $w_E \approx 0.03 - 0.1$.

**General details.**  In most cases, we use the StepLR learning rate scheduler, sometimes with a linear warmup. For the cases of the HME21 and water datasets, we use slightly more complicated schemes that are described in the corresponding paragraphs. Unless otherwise specified, we use $d_{\mathrm{PET}} = 128$, $n_{GNN} = n_{TL} = 3$, multi-head attention with 4 heads, SiLU activation, and the dimension of the feed-forward network model in the transformer is set to 512. We use a rotational augmentation strategy during fitting, which involves randomly rotating all samples at each epoch. This is accomplished using Scipy's implementation[98] of a generator that provides uniformly distributed random rotations. We employ Adam[99] optimizer for all cases. For all models with $d_{\mathrm{PET}} = 128$, we set the initial learning rate at $10^{-4}$. However, some models with $d_{\mathrm{PET}} = 256$ were unstable at this learning rate. Consequently, we adjusted the initial learning rate to $5 \cdot 10^{-5}$ for two cases: 1) $CH_4$ E+F, 100k samples, and 2) the COLL dataset. We never use dropout or weight decay, but rather avoid overfitting using an early-stopping criterion on a validation set.

# C  Detailed benchmark results

## C.1  COLL

The COLL dataset contains both "energy" and "atomization_energy" targets, that can be used together with the force data as gradients, and that are entirely equivalent after subtracting self-contributions. We follow the general training procedure summarized above. The most noteworthy detail is that COLL is the only dataset for which we observed molecular geometries for which all neighbor pairs lead to degenerate, collinear coordinate systems. We tackle this problem by using as a fallback a modified version of PET as an internally rotationally invariant model, see Appendix F.3 for details. In order to enforce rotational invariance in PET we modify the encoding procedure of the geometric information into input tokens. We encode only information about the length of the displacement vector $r_{ij}$, as opposed to the entire displacement vector $\mathbf{r}_{ij}$, as in the standard PET. As a result, the model becomes intrinsically rotationally invariant but loses some of its expressive power: the rotationally-invariant PET belongs to the class of 2-body GNNs, that have been shown to be incapable of discriminating between specific types of point clouds, see Ref. 32. Nevertheless, the accuracy of the 2-body PET remains surprisingly high. Furthermore, this auxiliary model was active only for 85 out of 9480 test molecules, while all other structures were handled completely by the main PET-256 model. Therefore, the accuracy of the auxiliary model has a negligible impact on the accuracy of the overall model. We summarize the performance of several models in Table 2.

Table 2: Comparison of the accuracy of different versions of PET on the COLL dataset, including also the models reported in Ref. [17]. PET-256 is the main model with $d_{PET} = 256$. PET-128-2-body is the auxiliary internally invariant model with $d_{PET} = 128$. PET-128 is a normal PET model $d_{PET} = 128$ and all the other settings match the ones of PET-128-2-body. PET-ECSE is an overall rotationally invariant model constructed with PET-256 as the main model and PET-128-2-body as the auxiliary one. The accuracy of SchNet is reported in Ref. [17].

| model | MAE $f$, meV/Å | MAE $E$, meV/molecule |
|---|---|---|
| SchNet[31] | 172 | 198 |
| DimeNet$^{++}$[86] | 40 | 47 |
| GemNet[17] | 26.4 | 53 |
| PET-128-2-body | 56.7 | 27.4 |
| PET-128 | 29.8 | 15.3 |
| PET-256 | 23.1 | 12.0 |
| PET-ECSE | 23.1 | 11.9 |

## C.2 HME21

For the HME21 dataset we employ a learning rate scheduler that differs from the standard StepLR. As shown in Fig. C.2, when StepLR reduces the learning rate, one observes a rapid decrease of validation error, which is however followed by clear signs of overfitting. To avoid this, we implement an alternative scheduler that decreases the learning rate rapidly to almost zero. As shown in Fig. C.2 (orange lines) this alternative scheduler leads to a noticeable improvement in validation accuracy.

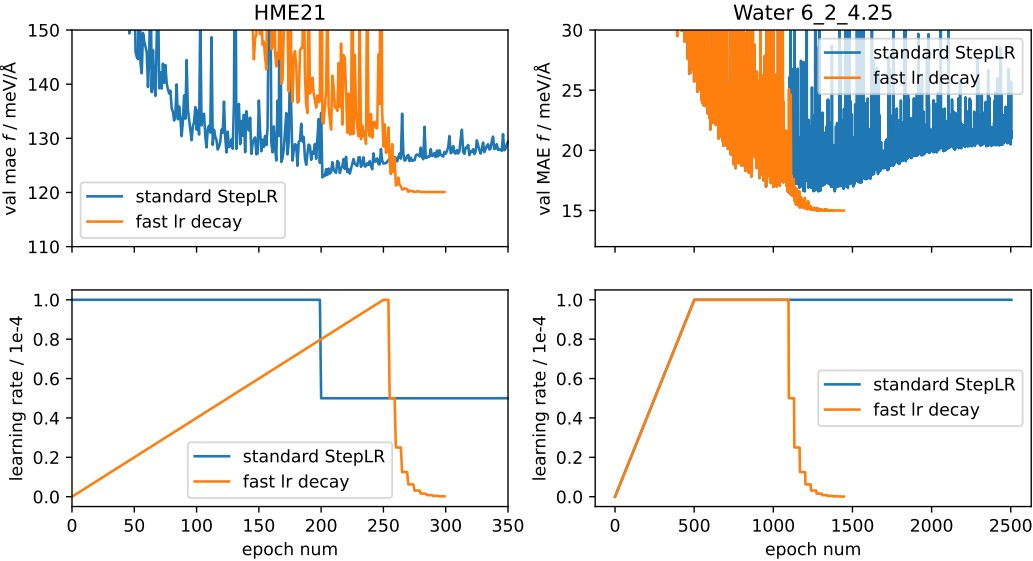

Figure 4: Comparison of a standard StepLR learning rate schedule and the strategy we use for the HME21 and water datasets.

In addition, we slightly modify the loss function for this dataset. The error metric for energies used in prior work is the Mean Absolute Error (MAE) per atom, computed first by determining energies per atom for both the ground truth energies and predictions, then by calculating the MAE for these normalized values. Therefore, it is logical to adjust the part of the loss function related to energies to reflect this metric. We re-define the loss as the mean squared error between the ground truth energies and predictions per atom. This is equivalent to the weighted loss function where weights are given by the inverse of the number of atoms in the structure. We observe that the performance of the model fitted with such a modified loss is nearly identical to that obtained when targeting total energies.

Table C.2 compares the results of PET models with those of recent approaches from the literature. Besides the ECSE-symmetrized PET model, and the non-symmetrized results obtained with a single evaluation (which are also computed for other datasets) here we also investigate the impact of test rotational augmentation, which appears to slighlty improve the test accuracy.

Table 3: A comparison between the accuracy of PET and that of SchNet, TeaNet, PaiNN and NequIP (values reproduced from Ref. [100]) and MACE [89]. We compute 5 different PET models, and report both the mean errors (including also the standard deviation between the models) as well as the error on the ensemble models obtained by averaging the predictions of the 5 models. The regularizing effect of ensemble averaging leads to a noticeable improvement in accuracy. We also compare the results from a the non-symmetrized model (PET-1), those obtained by averaging 10 random orientations (PET-10) and those with ECSE applied upon inference.

| model | MAE $E$, meV/at. | MAE $\|f\|$, meV/Å | MAE $f$, meV/Å |
|---|---|---|---|
| MACE[89] | 15.7 | 138 | |
| TeaNet[101] | 19.6 | 174 | 153 |
| SchNet[31] | 33.6 | 283 | 247 |
| PaiNN[18] | 22.9 | 237 | 208 |
| NequIP[35] | 47.8 | 199 | 175 |
| PET-1 | $17.8 \pm 0.1$ | $140.5 \pm 2.0$ | $124.0 \pm 1.6$ |
| PET-1-ens | 16.8 | 128.1 | 113.6 |
| PET-10 | $17.7 \pm 0.1$ | $139.9 \pm 1.9$ | $123.5 \pm 1.6$ |
| PET-10-ens | 16.8 | 127.9 | 113.5 |
| PET-ECSE | $17.8 \pm 0.1$ | $141.6 \pm 1.9$ | $125.1 \pm 1.6$ |
| PET-ECSE-ens | 16.8 | 128.5 | 114 |

## C.3 MnO

We randomly split the dataset into a training subset of 2608 structures, a validation subset of 200 structures, and a testing subset of 293 structures. Thus, the total size of our training and validation subsets matches the size of the training subset in Ref. [88]. One should note that we use all forces in each epoch, while Ref. [88] only uses a fraction of the forces – typically those with the largest error. Given that the selected forces can change at each epoch, however, all forces can be, at one point, included in the training.

In addition to the position of each atom in 3D space and a label with atomic species, this dataset contains real values of collinear magnetic moments associated with each atom. The continuous values of the atomic spins are obtained by a self-consistent, spin-polarized density-functional theory calculation. Even though using this real-valued atomic spins as atomic labels improves substantially the accuracy of the PET model, these are quantities that cannot be inferred from the atomic positions, and therefore it is impractical to use them for inference. Similar to what is done in Ref. 88, we discretize the local spin values, rounding them to the according to the rule $(-\infty, -0.25] \rightarrow -1$, $(-0.25, 0.25) \rightarrow 0$, $[0.25, \infty) \rightarrow 1$. These discrete values can be sampled, e.g. by Monte Carlo, to find the most stable spin configuration, or to average over multiple states in a calculation. We also report the error in energies of a model that doesn't use spins at all, which agrees within two significant digits with the analogous results reported in Ref. [88]. This suggests that both models were able to extract all possible energetic information from an incomplete description that disregards magnetism.

## C.4 High-entropy alloys

The main dataset of High-Entropy-Alloy configurations, which we used for training, was generated by randomly assigning atom types chosen between 25 transition metals to lattice sites of fcc or bcc crystal structures, followed by random distortions of their positions[102]. The resulting configurations were then randomly shuffled into training, validation, and testing datasets, following a protocol similar to that discussed in Ref. 90. We used 24630 samples for training, 500 for validation, and 500 for testing. While Ref. 90 uses all the energies available for the training dataset, we acknowledge that it uses only part of the forces. This, however, can be attributed primarily to the considerable Random

Table 4: Comparison between the accuracy of the high-dimensional neural network model of Ref. 88, and that of several (ECSE-symmetrized) PET models. Spins-discretized models approximate the atomic spins to integer values, PET-ECSE-spins uses the real-valued spins as atomic labels, while PET-ECSE-no-spins treats all Mn atoms as if they had no magnetization.

| model | MAE $E$, meV | MAE $f$, meV/Å |
|---|---|---|
| mHDNNP-spins-discretized[88] | 1.11 | 125 |
| PET-ECSE-spins-discretized | 0.30 | 22.7 |
| PET-ECSE-spins | 0.14 | 8.5 |
| PET-ECSE-no-spins | 11.6 | 94.8 |

Access Memory (RAM) demands imposed by the model introduced in Ref.90. The efficiency of PET allowed us to use all the available forces on a standard node of our GPU cluster.

In addition, Ref. 90 proposes additional "out-of-sample" test scenarios. In particular, it provides molecular-dynamics trajectories of a configuration containing all 25 atom types, performed at temperatures of $T = 300K$ and $T = 5000K$. For the trajectory at $T = 300K$, the errors of the PET model (ECSE-symmetrized) are 9.5 meV/atom and 0.13 eV/Å for energies and forces, respectively, which compares favorably with the values of 14 meV/atom and 0.23 eV/Å obtained with the HEA25-4-NN model. For the $T = 5000K$ trajectory, which undergoes melting and therefore differs substantially from the crystalline structures the model is trained on, PET yields errors of 152 meV/atom and 0.28 eV/Å. HEA25-4-NN yelds lower energy error and comparable force error, 48 meV/atom and 0.29 eV/Å respectively.

## C.5  Water

We randomly split the dataset[103] into a training subset of size 1303, a validation subset of size 100, and a testing subset of size 190. For the water dataset, we apply a similar trick with fast learning rate decay as we discussed for HME21 abpve. We start fast decaying of the learning rate at the epoch with the best validation error. The right subplot of Fig. C.2 illustrates the improvement achieved by this strategy compared to the constant learning rate for the model with $n_{GNN} = 6$, $n_{TL} = 2$ and $R_c = 4.25$Å. We use this trick for the ablation studies illustrated in panels b and c of Fig. 3 in the main text and not for those depicted in panel a.

## C.6  CH$_4$

This dataset contains more than 7 million configurations of a CH$_4$ molecule, whose atomic positions are randomized within a 3.5Å sphere, discarding configurations where two atoms would be closer than 0.5Å[104]. Our testing subset consists of configurations with indices from 3050000 to 3130000 in the original XYZ file. For the validation set, we use indices from 3000000 to 3005000. When we train a model with $n_{train}$ training configurations, we use the ones with indices from 0 to $n_{train}$. These settings match those of Ref. [80]. We use a full validation dataset for fitting the $d_{PET} = 256$ models and a 5k subset for all the $d_{PET} = 128$ ones in order to reduce the computational cost.

For methane, the only metric of interest is the Root Mean Square Error (RMSE) in energies, even when the fitting involves both energies and forces. The loss function we use is defined in equation (8). This definition includes the dimensionless hyperparameter $w_E$, which determines the relative importance of energies. We have observed that, surprisingly, the best accuracy in energies is achieved when $w_E \ll 1$. For all our training runs, we used $w_E = 0.125$.

## C.7  QM9 energies

We randomly split the dataset into a training subset of size 110000, a validation subset of size 10000, and a testing subset of size 10831. The reported MAE of 6.7 meV/molecule corresponds to the base model, without application of the ECSE protocol. Table 5 compares the accuracy of PET with several other models, including the current state-of-the-art model, Wigner Kernels[81].

Table 5: A comparison of the accuracy of several recent models for predicting the atomization energy of molecules in the QM9 dataset.

| Model | Test MAE (meV) |
|---|---|
| Cormorant[105] | 22 |
| Schnet[31] | 14 |
| EGNN[106] | 11 |
| NoisyNodes[107] | 7.3 |
| SphereNet[108] | 6.3 |
| DimeNet++[86] | 6.3 |
| ET[109] | 6.2 |
| PaiNN[18] | 5.9 |
| Allegro[72] | $4.7 \pm 0.2$ |
| WK[81] | $4.3 \pm 0.1$ |
| PET | 6.7 |

## C.8 QM9 dipoles

The dataset from Ref. [82] contains *vectorial* dipole moments computed for 21000 molecules from the QM9 dataset. These are not to be confused with the scalar values of the norms of dipole moments contained in the original QM9 dataset. We use 1000 configurations as a testing subset, similarly to WK[81]. Our validation subset contains 500 molecules, and we use up to 19500 samples for training.

## D  Reproducibility Statements

In order to facilitate the reproducibility of the results we present, we have released the following assets: 1) the source code for the PET model, 2) the source code for our proof-of-principle implementation of the ECSE protocol, 3) a complete set of hyperparameters for each training procedure, organized as a collection of YAML files, 4) similarly organized hyperparameters for the ECSE, 5) the Singularity container used for most numerical experiments 6) all the checkpoints, including those obtained at intermediate stages of the training procedure, and 7) the datasets we used. This should suffice for the reproducibility of all experiments reported in this manuscript. All these files are available at:

`https://doi.org/10.5281/zenodo.7967079`

In addition, the most important details of the different models we trained are discussed in Appendix B.

## E  Computational resources

Training times substantially depend on the dataset used for fitting. For instance, fitting the PET model with $d_{PET} = 128$ on the $CH_4$ dataset using 1000 energy-only training samples (the very first point of the learning curve in Fig. 3d) takes about 40 minutes on a V100 GPU. In contrast, fitting the PET model with $d_{PET} = 256$ on energies and forces using 300,000 samples is estimated to take about 26 GPU-days on a V100 (in practice the model was fitted partially on a V100 and partially on an RTX-4090). A similar computational effort is associated with training the $d_{PET} = 256$ model on the COLL dataset. It is worth mentioning, however, that the model already improves upon the previous state-of-the-art model in forces after approximately one-third of the total fitting time on the *validation* dataset. We used $d_{PET} = 256$ models only for 3 cases: 1) $CH_4$ E+F, 100k samples, 2) $CH_4$ E+F, 300k samples, and 3) COLL dataset. All the other calculations were substantially cheaper. For example, using a V100 GPU it takes about 30h to achieve the best validation error on the HME21 dataset for a single $d_{PET} = 128$ model.

## F  ECSE

The overall idea behind the ECSE symmetrization protocol is relatively simple, but an efficient implementation requires a rather complicated construction, that we discuss here in detail.

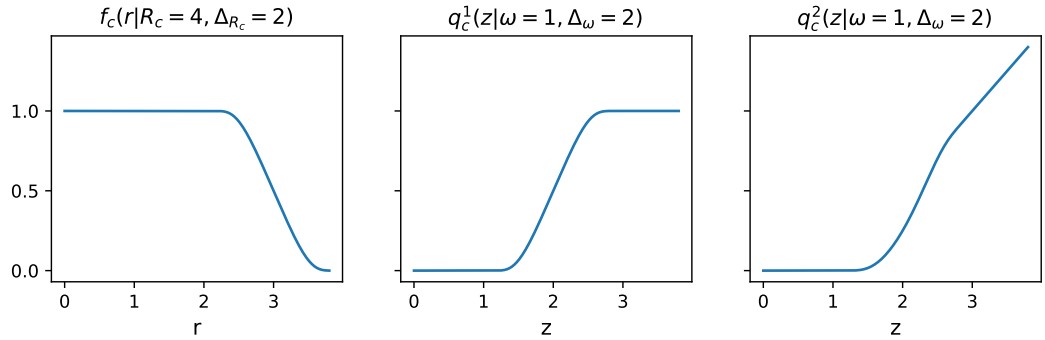

Figure 5: Smooth cutoff functions $f_c$, $q_c^1$ and $q_c^2$.

## F.1 Preliminary definitions

We begin by defining several mathematical functions that we use in what follows. In order to obtain smooth predictions, PET uses several cutoff functions. While there is a degree of flexibility in defining them, we found the following functional forms, that are also illustrated in Fig. 5, to be effective:

$$f_c(r|R_\text{c}, \Delta_{R_c}) = \begin{cases} 1, & \text{if } r \leq R_\text{c} - \Delta_{R_c} \\ \frac{1}{2}\left(\tanh\left(\frac{1}{x+1} + \frac{1}{x-1}\right) + 1\right); x = 2\frac{r - R_c + 0.5\Delta_{R_c}}{\Delta_{R_c}}, & \text{if } R_\text{c} - \Delta_{R_c} < r < R_\text{c} \\ 0, & \text{if } r \geq R_\text{c} \end{cases}$$
(9)

$$q_c^1(z|\omega, \Delta_\omega) = \begin{cases} 0, & \text{if } z \leq \omega \\ \frac{1}{2}\left(-\tanh\left(\frac{1}{x+1} + \frac{1}{x-1}\right) + 1\right); x = 2\frac{z - \omega - 0.5\Delta_\omega}{\Delta_\omega} & \text{if } \omega < z < \omega + \Delta_\omega \\ 1, & \text{if } z \geq \omega + \Delta_\omega \end{cases} \quad (10)$$

$$q_c^2(z|\omega, \Delta_\omega) = \begin{cases} 0, & \text{if } z \leq \omega \\ \frac{z}{2}\left(-\tanh\left(\frac{1}{x+1} + \frac{1}{x-1}\right) + 1\right); x = 2\frac{z - \omega - 0.5\Delta_\omega}{\Delta_\omega} & \text{if } \omega < z < \omega + \Delta_\omega \\ z, & \text{if } z \geq \omega + \Delta_\omega \end{cases} \quad (11)$$

We also need smooth functions to prune the ensemble of coordinate systems. To do so, we use several variations on a well-known smooth approximation for the *max* function. Given a finite set of numbers $\{x_i\}_i$ and a smoothness parameter $\beta > 0$ one can define:

$$\text{SmoothMax}(\{x_i\}_i|\beta) = \frac{\sum_i \exp(\beta x_i) x_i}{\sum_i \exp(\beta x_i)}. \tag{12}$$

This function satisfies the following properties:

$$\begin{aligned} \text{SmoothMax}(\{x_i\}_i|\beta) &\leq \max(\{x_i\}_i) \\ \lim_{\beta \to +\infty} \text{SmoothMax}(\{x_i\}_i|\beta) &= \max(\{x_i\}_i) \end{aligned} \tag{13}$$

For a set of just two numbers $\{x_1, x_2\}$ one can show that SmoothMax is bounded from below:

$$\begin{aligned} \text{SmoothMax}(\{x_1, x_2\}) + T(\beta) &\geq \max(\{x_1, x_2\}) \\ \lim_{\beta \to +\infty} T(\beta) &= 0 \end{aligned} \tag{14}$$

for $T(\beta) = W(\exp(-1))/\beta$, where $W$ is the Lambert W function.

For the case when the numbers $x_i$ are associated with weights $p_i \geq 0$ it is possible to extend the definition of SmoothMax as:

$$\text{SmoothMaxWeighted}(\{(x_i, p_i)\}_i|\beta) = \frac{\sum_i \exp(\beta x_i) p_i x_i}{\sum_i \exp(\beta x_i) p_i}. \tag{15}$$

This function satisfies the properties in Eq. (13), where maximum is taken only of the subset of $\{x_i\}$ where $p_i > 0$. A useful consequence of the definition of *SmoothMaxWeighted* is that its value is not changed when including a point with zero weight, $(x, 0)$:

$$\text{SmoothMaxWeighted}(\{(x_i, p_i)\}_i \cup \{(x, 0)\}|\beta) = \text{SmoothMaxWeighted}(\{(x_i, p_i)\}_i|\beta) \quad (16)$$

Corresponding functions that provide a smooth approximation to the $\min$ operation, *SmoothMin* and *SmoothMinWeighted*, can be defined the same way as *SmoothMax* and *SmoothMaxWeighted* using $(-\beta)$ instead of $\beta$.

Finally, given an ordered pair of noncollinear unit vectors $\hat{v}_1$ and $\hat{v}_2$ we define the associated coordinate system by computing $\mathbf{u}_1 = \hat{v}_1 \times \hat{v}_2$, $\hat{u}_1 = \mathbf{u}_1/\|\mathbf{u}_1\|$ and $\hat{u}_2 = \hat{v}_1 \times \hat{u}_1$. The reference frame is given by the vectors $\hat{v}_1$, $\hat{u}_1$ and $\hat{u}_2$. This procedure generates only right-handed coordinate systems, and thus, the transformation between the absolute coordinate system to the atom-centered one can always be expressed as a proper rotation, never requiring inversion. When we later refer to the coordinate system given by a central atom $i$ and two neighbors $j$ and $j'$, we mean that the vectors $\hat{v}_1$ and $\hat{v}_2$ are the ones pointing from the central atom to neighbors $j$ and $j'$ respectively.

## F.2 Assumptions

We assume that (1) there is a finite number of neighbors within any finite cutoff sphere and (2) there is a lower boundary $d_{\min}$ to the distance between pairs of points. Neither of these assumptions is restrictive in chemical applications, because molecules and materials have a finite atom density, and because atoms cannot overlap when working on a relevant energy scale.

## F.3 Fully-collinear problem

The simplest version of ECSE protocol was defined in the main text in the following way:

$$\mathbf{y}_S(A_i) = \sum_{jj' \in A_i} w_{jj'} \hat{R}_{jj'} [\mathbf{y}_0(\hat{R}_{jj'}^{-1}[A_i])] \Big/ \sum_{jj' \in A_i} w_{jj'}, \quad (17)$$

where

$$w_{jj'} = w(\mathbf{r}_j, \mathbf{r}_{j'}) = f_c(r_j|R_c, \Delta_{R_c}) f_c(r_{j'}|R_c, \Delta_{R_c}) q_c(|\hat{r}_j \times \hat{r}_{j'}|^2|\omega, \Delta_\omega). \quad (18)$$

The $q_c$ function can be chosen as either $q_c^1$ or $q_c^2$, while $\omega$ and $\Delta_\omega$ are user-specified constant parameters.

This simple formulation doesn't handle correctly the corner case of when all pairs of neighbors form nearly collinear triplets with the central atom. In this scenario, for all pairs of $j$ and $j'$, $|\hat{r}_j \times \hat{r}_{j'}|^2$ is smaller than $\omega$ which implies that all the angular cutoff values $q_c(|\hat{r}_j \times \hat{r}_{j'}|^2|\omega, \Delta_\omega)$ are zeros, and so all the weights $w_{jj'}$. As a result, the ECSE weighted average is undefined, falling to $\frac{0}{0}$ ambiguity. We propose two solutions to treat this case.

**First solution to the fully-collinear problem.** The first solution we propose is to incorporate predictions of some internally equivariant model in the functional form of ECSE:

$$\mathbf{y}_S(A_i) = \left( w_{\text{aux}} \mathbf{y}_{\text{aux}}(A_i) + \sum_{jj' \in A_i} w_{jj'} \hat{R}_{jj'} [\mathbf{y}_0(\hat{R}_{jj'}^{-1}[A_i])] \right) \Big/ \left( w_{\text{aux}} + \sum_{jj' \in A_i} w_{jj'} \right), \quad (19)$$

In this case, $\mathbf{y}_S(A_i)$ can fall to $\mathbf{y}_{\text{aux}}(A_i)$ if all the weights $w_{jj'}$ are zeros.

Under the assumption that the (typically simpler) auxiliary model will be less accurate than the main backbone architecture, it is desirable to ensure that it is only used to make predictions for corner cases. For this purpose we define $w_{\text{aux}}$ as follows:

$$w_{\text{aux}} = f_c(\text{SmoothMaxWeighted}(\{w_{jj'}, w_{jj'}\}_{jj'}|\beta_\omega)|t_{\text{aux}}, \Delta_{\text{aux}}). \quad (20)$$

With this expression, $w_{\text{aux}}$ takes a non-zero weight only when the smooth maximum of the coordinate-system weights $w_{jj'}$ is below $t_{\text{aux}}$: in other words, as long as at least one pair of neighbors is non-collinear, the auxiliary model will be ignored. Since *SmoothMaxWeighted* does not depend on quantities with zero weight (see Eq. (16)), one can compute Eq. (20) efficiently, without explicit iterations over the coordinate systems with zero weight. $\beta_\omega$, $t_{\text{aux}}$ and $\Delta_{\text{aux}}$ are user-specified constant parameters.

**Second solution to the fully-collinear problem.** Even though the use of an auxiliary model is an effective and relatively simple solution, it would be however desirable to use consistently the non-equivariant backbone architecture. In this section, we discuss a sketch of how one could tackle this corner-case without the usage of an auxiliary model.

To eliminate the $0/0$ singularity, we need to use an adaptive definition for the angular cutoff parameters $\omega$ and $\Delta_\omega$ that enter the definition (18) of the ECSE weights:

$$\omega = \Delta_\omega = \frac{1}{2} \mathrm{SmoothMax}(\{(|\hat{r}_j \times \hat{r}_{j'}|^2)\}_{jj'}|\beta). \tag{21}$$

With this definition, the weights are never zero except for the exact fully-collinear singularity, which makes the weighted average of the ECSE protocol well-defined everywhere. For environments approaching the exact fully-collinear singularity, the coordinate systems used by ECSE are not stable. However, given the definition of the coordinate systems discussed in Appendix F.1, the $x$ axes of all the coordinate systems are aligned along the collinear direction. Given that the atomic environment itself is nearly collinear, the input to the non-equivariant models is always the same, irrespective of the $y$ and $z$ axes, that will be arbitrarily oriented in the orthogonal plane. Thus, all evaluations of the backbone architecture $\mathbf{y}_0$ lead to approximately the same predictions. Note that this is only true if all the neighbors that are used for the backbone architecture are also considered to build coordinate systems, which requires that the cutoff radius employed by ECSE is larger than that of the backbone architecture. When atoms approach smoothly a fully-collinear configuration, the coordinates that are input to the backbone architecture converge smoothly to being fully aligned along $x$ for all coordinate systems, making the ECSE average smooth.

One small detail is that, within this protocol, the *orientation* of the $x$ axis can be arbitrary. This problem is easily solved by always including an additional coordinate system in which $\hat{v}_1$ is oriented along $-\mathbf{r}_{ij}$ rather than along $\mathbf{r}_{ij}$. Both coordinate systems should be used with the same weight $w_{jj'}$. One should note, however, that this approach has a number of limitations. In particular, it doesn't support covariant inputs, such as vectorial spins associated with each atom, and covariant outputs, such as dipole moments.

## F.4 Adaptive inner cutoff radius

The minimal inner cutoff radius $R_{\mathrm{in}}$ should be chosen to ensure that it includes at least one pair of neighbors that define a "good" coordinate system. To determine an inner cutoff that contains at least a pair of neighbors, we proceed as follows

1. For each pair of neighbors, we use $\mathrm{SmoothMax}(r_j, r'_j|\beta)$ to select the farthest distance in each pair

2. Applying a *SmoothMin* function to the collection of neighbor pairs

$$\mathrm{SmoothMin}(\{\mathrm{SmoothMax}(\{r_j, r_{j'}\}|\beta) + T(\beta)\}_{jj'}|\beta) \tag{22}$$

   selects a distance that contains at least a pair of neighbors. That this function is always greater or equal to the second-closest pair distance follows from the inequality (14) and the *SmoothMin* analogue of Eq. (13).

3. We ensure to pick at least one non-collinear pair (if there is one within $R_{\mathrm{out}}$) by defining the adaptive inner cutoff as

$$R_{\mathrm{in}} = \mathrm{SmoothMinWeighted}(\{\mathrm{SmoothMax}(\{r_j, r_{j'}\}|\beta) + T(\beta), p_{jj'}\}_{jj'} \cup \{(R_{\mathrm{out}}, 1.0)\}|\beta) + \Delta_{R_c}, \tag{23}$$

   where $p_{jj'}$ are defined as:

$$p_{jj'} = f_c(r_j|R_{\mathrm{out}}, \Delta_{R_c}) f_c(r_{j'}|R_{\mathrm{out}}, \Delta_{R_c}) q_c^1(|\hat{r}_j \times \hat{r}_{j'}|^2|\omega + \Delta_\omega, \Delta_\omega). \tag{24}$$

   $j$ and $j'$ run over all the neighbors inside the outer cutoff sphere.

This definition maintains smoothness with respect to the (dis)appearance of atoms at the outer cutoff sphere. It ensures that if there's at least one good pair of neighbors inside the outer cutoff sphere, there's at least one within the inner cutoff as well. If no such good pair exists within the outer cutoff sphere, the definition falls back to $R_{\mathrm{out}}$. Note that the expression includes also some "tolerance" parameters $\Delta_{R_c}$ and $\Delta_\omega$. These are introduced to ensure that at least one of the weights computed to define the equivariant ensemble in Eq. (24) is of the order of 1.

## F.5 Weights pruning

Given that the cost of applying ECSE depends on the number of times the inner model needs to be evaluated, it is important to minimize the number of coordinate systems that have to be considered. To do so, one can e.g. disregard all coordinate systems with weights below a threshold, e.g. half of the maximum weight. A smooth version of this idea can be implemented as follows:

---

**Constant parameters:**
$\beta_w : \beta_w > 0$
$T_f, \Delta_{T_f}: T_f > 0, \Delta_{T_f} > 0, T_f < 1$
$T_e, \Delta_{T_e}: T_e > 0, \Delta_{T_e} > 0, \Delta_{T_e} < T_e$ ($T$ stands for Threshold).
**Smooth weights pruning:**
  1: $w_{\text{MAX}} \leftarrow \text{SmoothMaxWeighted}(\{(w_{jj'}, w_{jj'})\}_{jj'}|\beta_w)$
  2: $f_{jj'} \leftarrow q_c^1(w_{jj'}|w_{\text{MAX}}T_f, w_{\text{MAX}}\Delta_{T_f})$
  3: $e \leftarrow f_c(w_{\text{MAX}}|T_e, \Delta_{T_e})$
  4: $w_{jj'} \leftarrow ew_{jj'} + (1-e)w_{jj'}f_{jj'}$

---

First, one computes the maximum weight present, using a *SmoothMaxWeighted* function. Then, pruning factors $f_{jj'}$ are computed using the cutoff $q_c^1$ function, aiming to zero out the smallest weights with $w_{jj'} \leftarrow w_{jj'}f_{jj'}$. An additional modification is needed to ensure smooth behavior around fully-collinear configurations, where all weights become zero. If this occurs, $q_c^1(w_{jj'}|w_{\text{MAX}}T_f, w_{\text{MAX}}\Delta_{T_f})$ converges to a step function. Therefore, it makes sense to activate weight pruning only if $w_{\text{MAX}}$ is larger than a certain threshold $T_e$, which is implemented using a smooth activation factor $e$. We found this pruning to be more efficient if the $q_c$ function in Eq. (4) is selected to be $q_c^2$ rather than $q_c^1$. For multi-component systems, an alternative approach (which can be used on top of the discussed strategy) to increase efficiency involves using only a subset of all the atoms of specific atomic species to define coordinate systems (e.g., only O atoms in liquid water).

## F.6 Tradeoff between smoothness and computational efficiency

For any set of selected parameters, the ECSE protocol yields a smooth approximation of the target property in a rigorous mathematical sense, meaning the resulting approximation of the target property is continuously differentiable. The important question, however, pertains to the smoothness of this approximation in a physical sense. That is, how quickly it oscillates or, equivalently, how large the associated derivatives are. In this section, we discuss the tradeoff between "physical" smoothness and the computational efficiency of the ECSE protocol. This tradeoff is controlled by the user-specified parameters discussed in the previous paragraphs.

$\beta$ is an example of such parameters, with a clear impact on the regularity of equivariant predictions. Among other things, it is used to determine the inner cutoff radius, $R_{\text{in}}$, which is defined along the lines with a smooth approximation of the second minimum of all the distances from the central atom to all its neighbors. Assigning a relatively low value to $\beta$ generates a very smooth approximation of this second minimum, but it significantly overestimates the exact value, and therefore $R_{\text{in}}$ potentially includes more neighbor pairs than needed. In contrast, for a large value of $\beta$, the second minimum approximation reflects the exact value of the second minimum more precisely, at the cost of decreasing the smoothness of the approximation when the distances between atoms close to the center vary. Even though the ensemble average remains continuously differentiable for any $\beta > 0$, it might develop unphysical spikes in the derivatives, reducing its accuracy. To see more precisely the impact of $\beta$ consider the limit of a large value (considering also that other adjustment parameters such as $\Delta_{R_c}$ are set to a value that does not entail an additional smoothening effect). In this limit, the inner cutoff selected by the ECSE protocol encompasses only one pair of neighbors for a general position, so most of the time the ECSE protocol uses only one coordinate system defined by the two closest neighbors. Only when the nearest pair of atoms changes, within a narrow transition region, two coordinate systems are used simultaneously. The value of $\beta$ determines the characteristic size of this transition region. Thus, for excessively large values of $\beta$, there is a sharp (but still continuously differentiable) transition from one coordinate system to another. On the other hand, if $\beta$ is set to a very low value, $R_{\text{in}}$ will always be significantly larger than the exact distance to the second neighbor. As a result, the ECSE protocol will constantly utilize multiple coordinate systems, leading to a smoother symmetrized model but with a higher computational cost.

In summary, the parameters used by the ECSE dictate the balance between smoothness and computational efficiency. We name the selection of such parameters that leads to a very smooth model as "loose". The opposite selection that minimizes computational cost is labeled as "tight". The proper selection of these parameters directly impacts the accuracy of the symmetrized model in forces (and in higher derivatives), which we discuss in the next section.

### F.7 Accuracy of a-posteriori-ECSE-symmetrized models

The most computationally efficient method for fitting a model is to initially train a non-equivariant backbone architecture with random rotational augmentations and then apply the ECSE protocol a posteriori. There are general considerations one can make on the expected accuracy of the symmetrized model relative to the initial, non-equivariant one.

The functional form of ECSE is a weighted average of predictions generated by a backbone architecture for several coordinate systems. Therefore, it can be seen as implementing the rotational augmentation strategy during inference. This implies that the accuracy of ECSE in energies (and in any other direct targets) is expected to be bounded between the accuracy of standard single inference of the backbone architecture and that evaluated with fully-converged rotational augmentations.

Forces, however, present a more complicated situation. The forces produced by ECSE are not just linear combinations of the predictions given by the backbone architecture. This complexity stems from the fact that the coordinate systems and weights used by ECSE depend on atomic positions themselves. Since forces are defined by the negative gradient of energy, it is necessary to take the derivatives of Eq. (17) with respect to atomic positions to obtain the predictions of ECSE. Additional terms related to $\frac{\partial \hat{R}_{jj'}}{\partial \mathbf{r}_k}$ and $\frac{\partial w_{jj'}}{\partial \mathbf{r}_k}$ are not associated with the forces produced by a backbone architecture $\frac{\partial \mathbf{y}_0([A_i])}{\partial \mathbf{r}_k}$. Consequently, the error in forces of the final, symmetrized model may significantly exceed that of the backbone architecture. This issue is more pronounced for a tight selection of ECSE parameters and less pronounced for a loose one. In principle, this can be mitigated by fine-tuning the overall symmetrized model. In this work, we have selected loose parameters of the ECSE protocol sufficient to ensure that the difference in accuracy between the backbone architectures and symmetrized models is minuscule, even without fine-tuning, which results however in a more pronounced computational overhead.

### F.8 Additional rotational augmentations

In some cases, such as for the $CH_4$ configurations, the total number of all possible coordinate systems is very limited. However, using a larger number of rotational augmentations may still be beneficial during inference. For such cases, it's possible to incorporate additional augmentations into the ECSE protocol as follows:

$$\mathbf{y}_S(A_i) = \frac{\sum\limits_{jj' \in A_i} \sum\limits_{\hat{R}_{aug} \in S_{aug}} w_{jj'} \hat{R}_{jj'} \hat{R}_{aug}[\mathbf{y}_0(\hat{R}_{aug}^{-1}\hat{R}_{jj'}^{-1}[A_i])]}{N_{aug} \sum\limits_{jj' \in A_i} w_{jj'}}, \tag{25}$$

where $S_{aug}$ is a predefined constant set of random rotations $\hat{R}_{aug}$.

### F.9 Application of the ECSE protocol to long-range models

Several models capable of capturing long-range interactions have been developed within the atomistic machine-learning community. These models cannot be directly cast into the local energy decomposition discussed in Sec. 2. However, they are constructed as either pre- or post-processing operations for the local models. Thus, these schemes can also benefit from the ECSE protocol by making it possible to use some models developed for generic point clouds as a local engine.

An example of such models is LODE[110]. As a first step, it constructs a representation that is inspired by the electrostatic potential. This three-dimensional function, or a collection of three-dimensional functions for each atomic species, captures the long-range information. This global potential is then used to define local descriptors, that are combined with other short-range features to model a local energy decomposition. Thus, the ECSE protocol can be applied to LODE, or similar schemes, without any modifications.

Another example of a long-range model is given in Ref. [111]. This scheme applies local models to compute partial charges associated with each atom and later uses them to calculate a non-local electrostatic energy contribution. The ECSE protocol can ensure that these partial charges remain invariant with respect to rotations.

## F.10 Application of the ECSE protocol to message-passing schemes

As already discussed in the main text, message-passing schemes can also be considered within the framework of local models, with their receptive field playing the role of a cutoff. Thus, the ECSE protocol is directly applicable to them. Whereas this approach has linear scaling, it is very computationally expensive. As an example, consider a GNN with 2 message-passing layers. At the first layer, atom $A$ sends some message, denoted as $m^1_{AB}$, to atom $B$. At the second, atom $B$ sends messages $m^2_{BC}$ and $m^2_{BD}$ to atoms $C$ and $D$, respectively. The second layer additionaly computes predictions associated with atoms $C$ and $D$, given all the input messages. If one naively applies the ECSE protocol, it is first employed to get a rotationally invariant prediction associated with atom $C$ and then to get a prediction associated with atom D. Both calculations rely on the message $m^1_{AB}$, computed at the first layer of the GNN. The problem is that, in contrast to normal GNNs, this message cannot be reused to compute predictions associated with atoms $C$ and $D$. The coordinate systems defined by the ECSE protocol for atoms $C$ and $D$ do not match each other, and thus, the message $m^1_{AB}$ should be recomputed for each of them. The problem becomes progressively more severe as the depth of the GNN increases.

If the size of a finite system or the size of a unit cell is smaller than the receptive field of a GNN, one can accelerate and simplify the discussed approach by defining a global pool of coordinate systems. This can be achieved by applying the ECSE protocol to all atoms, subsequently unifying all the coordinate systems and associated weights. If one employs pruning techniques similar to the one described in Appendix F.5, this approach can become highly efficient, outperforming the naive one even if the size of the system exceeds the receptive field of the model. For finite systems, an even more efficient scheme can be designed by applying the ECSE protocol to a single point associated with the whole system, such as the center of mass.

The most computationally efficient way, however, is to apply the ECSE protocol layerwise, symmetrizing all the output messages at each layer. In this case, one must be especially careful with the fitting scheme. The standard scheme of first training a non-invariant model with random rotational augmentations and then applying the layerwise ECSE protocol a posteriori could lead to a significant drop in accuracy. The strategy of random rotational augmentations during training corresponds to applying random rotation to an entire atomic configuration. Thus, during training, a model can learn to encode some information about relative orientations into the messages. However, if one then applies the ECSE protocol layer by layer, the coordinate systems of the atom that sends a message do not match those of the receiving one. Consequently, the information about relative orientations can no longer be propagated.

This problem can be solved by using individual augmentation strategy that stands for rotating all atomic environments independently. In this case, the model can encode only rotationally invariant information into the messages. Thus, one can expect that a similar accuracy will be achieved if one enables a layerwise ECSE protocol a posteriori. Furthermore, it is possible to extend this approach to covariant messages. One can specify how the messages should transform with respect to rotations, for example, as vectors or tensors of rank $l$. Then, when performing individual rotation augmentations, one should explicitly transform the messages from the coordinate system of the sending atom to that of the receiving one. The layerwise ECSE protocol should also be modified accordingly.

To sum up, the efficient layerwise ECSE protocol supports covariant messages with any transformation rule. However, this transformation rule should be specified as part of the architecture. The global ECSE protocol is more powerful, as it allows the model to learn a behavior of the messages with respect to rotations on its own.

An interesting question that we put out of the scope of this paper is the behavior of the messages with respect to rotations for the layerwise scheme if the ECSE protocol is turned on immediately and symmetrized model is trained from the very beginning.

### F.11 Current implementation

Our current implementation uses a global pool of coordinate systems, as discussed in Appendix F.10. In addition, it implements most of the logic of the ECSE protocol using zero-dimensional PyTorch tensors, each containing only one scalar. Thus, it has the potential to be substantially accelerated. We want to reiterate that we have selected loose parameters for the ECSE protocol corresponding to the best accuracy and substantial computational cost (see more details in Appendices F.6 and F.7). Therefore, even with the current proof-of-principle implementation, ECSE can be substantially accelerated at the cost of a minuscule drop in accuracy.

### F.12 Numerical verification of smoothness

To verify that our implementation of the ECSE protocol indeed preserves the smoothness of the backbone architecture, we numerically analyzed alterations of the predictions with respect to random perturbations of the input. For this purpose, we selected the $CH_4$ dataset (discussed in the Appendix C) since it represents random configurations, thus covering most of the configurational space. For 100 molecules from this dataset, we applied 50 random Gaussian perturbations to all the positions of all the atoms with different amplitudes. For each perturbation, the corresponding change in the prediction of the ECSE symmetrized model was measured. As a backbone architecture, we used a PET model with hyperparameters similar to the one benchmarked in Appendix C. The parameters of ECSE were selected to be loose in a sense discussed in Appendix F.6.

Results are shown in Fig. 6. One can see that a small amplitude of the applied noise entails a minor alteration of the predictions. No instances were observed where a minor noise led to a significant change in the predictions of the ECSE symmetrized model.

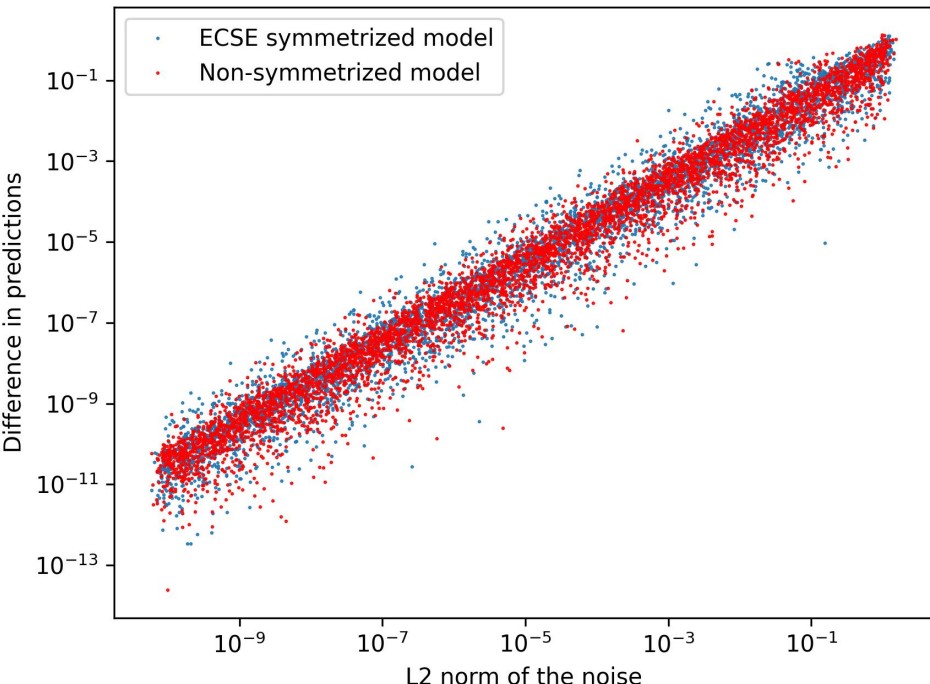

Figure 6: Numerical verification of preserving smoothness. Each point represents one random perturbation of one $CH_4$ molecule. The horizontal coordinate is an amplitude of Gaussian noise applied to all the positions of all the atoms. Vertical coordinate represents the difference in the predictions. The experiment was conducted for both symmetrized and non-symmetrized PET models.

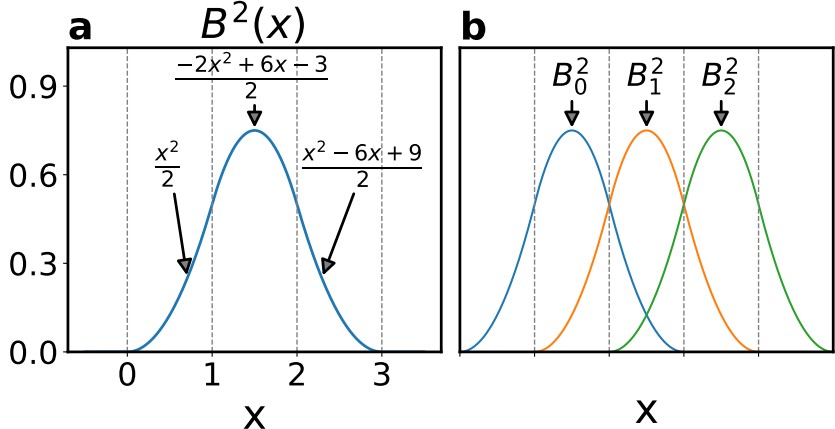

Figure 7: (a) B-spline of 2nd order. (b) Equidistant collection of B-splines. The figure is adapted from Ref. [112]

## G  Architectures for generic point clouds made smooth

In the main text, we stated that most architectures for generic point clouds can be made smooth (continuously differentiable) with relatively small modifications. Here we provide a few practical examples of how non-smooth point cloud architectures can be modified.

### G.1  Voxel and projection based methods

For a naive voxelization of a point cloud, voxel features are estimated based on the points inside the corresponding voxel. This means that this approach leads to discontinuities whenever a point moves from one voxel to another. The construction used in GTTP[112] and UF[113] is an example of a smooth projection of a point cloud onto the voxel grid. We first describe the smooth projection onto a one-dimensional pixel/voxel grid for simplicity. It relies on smooth, piecewise polynomial, bell-like functions known as B-splines.

Fig. 7a demonstrates the shape of a second-order B-spline. It spreads over three one-dimensional voxels, on each of them it is defined by a second-order polynomial, and is continuously differentiable. For any $p$, it is possible to construct an analogous function $B^p(x)$ which spreads over $p+1$ voxels, is a piecewise polynomial of order $p$, and is $p-1$ times continuously differentiable. Next, an equidistant collection of such B-splines is constructed, as shown in Fig. 7 (b). The projection onto the one-dimensional voxel grid is defined as follows:

$$c_i = \sum_k B_i^p(x_k), \tag{26}$$

where $x_k$ is the position of a k-th point. The coefficients $c_i$ can be treated as features associated with the i-th voxel.

Three-dimensional B-splines can be built as a tensor product of these functions, i.e.:

$$B_{i_1 i_2 i_3}^p(x, y, z) = B_{i_1}^p(x) B_{i_2}^p(y) B_{i_3}^p(z), \tag{27}$$

and the corresponding projection of a point cloud onto a 3D voxel grid is:

$$c_{i_1 i_2 i_3} = \sum_k B_{i_1 i_2 i_3}^p(\mathbf{r}_k). \tag{28}$$

An alternative method to define a smooth projection is to define a smooth point density, and to compute its integrals for each voxel, as demonstrated in Fig. 8 for the one-dimensional case. The point density is defined as follows:

$$\rho(\mathbf{r}) = \sum_k B(\mathbf{r} - \mathbf{r}_k), \tag{29}$$

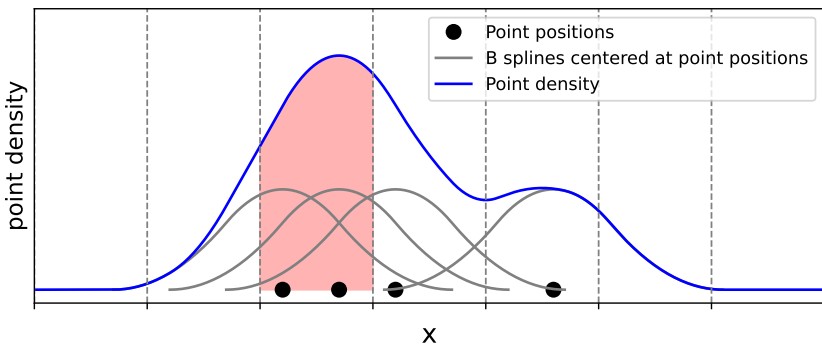

Figure 8: Integral projection. The integral of the red area determines the features of the third left voxel.

where $B$ is any bell-like function. In contrast to the previous example, here, bell-like functions are associated with each point rather than with each voxel.

The integral can be rewritten into the form of the first projection method:

$$\int_{\substack{\text{voxel} \\ i_1 i_2 i_3}} \rho(\mathbf{r}) d\mathbf{r} = \sum_k B^{\text{int}}_{i_1 i_2 i_3}(\mathbf{r}_k), \tag{30}$$

where

$$B^{\text{int}}_{i_1 i_2 i_3}(\mathbf{r}_k) = \int_{\substack{\text{voxel} \\ i_1 i_2 i_3}} B(\mathbf{r} - \mathbf{r}_k) d\mathbf{r}. \tag{31}$$

If $B$ is given by a B-spline of the $p$-th order, then $B^{\text{int}}$ is given by a B-spline of $p+1$-th order[114].

Since the B-spline functions have compact support, the discussed projections preserve the sparsity of the voxel features.

Within a paradigm of local energy decomposition, one can center the voxel grid at the position of the central atom. When building a projection of an atomic environment onto a local voxel grid, one last source of discontinuity is associated with the (dis)appearance of new atoms at the cutoff sphere. This problem can be avoided by modifying Eq. (28) as:

$$c_{i_1 i_2 i_3} = \sum_k f_c(r_k | R_c, \Delta_{R_c}) B^p_{i_1 i_2 i_3}(\mathbf{r}_k), \tag{32}$$

and Eq. (29) as:

$$\rho(\mathbf{r}) = \sum_k f_c(r_k | R_c, \Delta_{R_c}) B(\mathbf{r} - \mathbf{r}_k). \tag{33}$$

($\mathbf{r}_k$ here denotes the displacement vector from central atom to the $k$-th neighbor). Finally, the 3D convolutional NN applied on top of the voxel projection should be smooth. One can ensure this by selecting a smooth activation function and using smooth pooling layers, such as sum or average, or using a CNN without pooling layers. The ECSE protocol applied on top of this construction adds the only missing ingredient – rotational equivariance – which makes voxel-based methods applicable for atomistic modeling. The paradigm of local energy decomposition is not the only way to use voxel-based methods. For instance, for finite systems of moderate sizes, such as molecules and nanoparticles, one can center the voxel grid at the center of mass and apply ECSE protocol to the same point. Although intuitively, projection-based 2D methods are not expected to perform well given that the 3D information is crucial for estimating the energy of an atomic configuration, in principle, they can be made smooth and applied using the 2D analogue of the methods discussed above.

## G.2 Point models

A common building block of various point methods is the following functional form [48, 115]:

$$\mathbf{x}'_i = \gamma(\square(\{h(\mathbf{x}_i, \mathbf{x}_j) | j \in A_i\})), \tag{34}$$

where $\mathbf{x}_i$ is a feature vector associated with a central point $i$, $\mathbf{x}_j$ are feature vectors associated with the neighbors, $\gamma$ and $h$ are learnable functions, $\square$ is a permutationally invariant aggregation function, and the neighborhood $A_i$ is defined either as the $k$ nearest neighbors or as all the points within certain cutoff radius $R_c$. The distance used to define local neighborhoods can be computed either in the initial space or between the feature vectors $\mathbf{x}$ at the current layer of the neural network.

The construction above contains several sources of discontinuities even for smooth functions $\gamma$ and $h$. First, the definition of the local neighborhood as the $k$ nearest neighbors is not smooth with respect to the change of a set of neighbors; that is, when one point is leaving the local neighborhood, and another is entering. Thus, one should instead use the definition of a local neighborhood with a fixed cutoff radius. In principle, one can make it adaptive, meaning it could be a smooth function of a neighborhood, embracing $k$ neighbors on average.

If $\square$ is given by summation, then, as it was already outlined in the main text for point convolutions, the smooth form can be constructed as:

$$\mathbf{x}'_i = \gamma(\sum_j f_c(d_{ij}|R_c, \Delta_{R_c}) h(\mathbf{x}_i, \mathbf{x}_j)), \tag{35}$$

where $d_{ij}$ is the distance between points $i$ and $j$.

If $\square$ is given by a maximum, then one can ensure smoothness using an expression such as:

$$\mathbf{x}'_i = \gamma(\text{SmoothMax}(\{f_c^{\max}(d_{ij}|R_c, \Delta_{R_c}) h(\mathbf{x}_i, \mathbf{x}_j) | j \in \text{neighborhood}(\mathbf{x}_i)\})), \tag{36}$$

where $f_c^{max}(d_{ij}|R_c, \Delta_{R_c})$ is some smooth function which converges to $-\infty$ at $R_c$. Minimum and average aggregations can be adapted analogously.

We are unaware of any simple modification to ensure smoothness in the downsampling operations that are applied in PointNet++-like methods, and are typically performed by the Farthest Point Sampling algorithm. However, this operation is probably unnecessary for architectures designed for atomistic modeling due to the locality [116] of quantum mechanical interactions and the fact that many successful domain-specific models do not use anything resembling a downsampling operation.

To sum up, many neural networks developed for generic point clouds can be straightforwardly made smooth and, thus, can be made applicable to materials modeling if using the ECSE protocol. For another large group of architectures, we are currently unaware of how to ensure smoothness for *entire* models because of operations such as downsampling. However, in many such cases, core building blocks, or neural network layers, can be made smooth and used to construct PointNet-like architectures for atomistic applications.

# H  Broader impact

Machine learning force fields can significantly accelerate progress in areas such as drug discovery and materials modeling by alleviating the substantial computational costs associated with ab-initio simulations. Even though training models involves a substantial energy cost, the energy cost of inference is minuscule in comparison with that of the reference electronic structure calculations, which leads to overall energy savings. Although these models could theoretically be used for malicious intentions, we argue that the likelihood is minimal. This stems from the fact that machine learning force fields are not direct agents of impact. Instead, they function primarily as scientific tools, typically deployed within the framework of organized research institutions rather than being exploited by single individuals with malevolent objectives.

