# OpenReview forum: "Smooth, exact rotational symmetrization for deep learning on point clouds"
_NeurIPS.cc/2023/Conference — NeurIPS 2023 poster_

### Official Review · Reviewer_oCKF · 2023-07-02

**Soundness:** 3 good
**Presentation:** 3 good
**Contribution:** 2 fair
**Rating:** 4
**Confidence:** 3

**Summary:**

The authors propose a method for uniting rotational symmetries with translational and permutation symmetries in neural networks for point cloud data. Hereby, the authors particularly focus on molecule data. This is achieved by adding rotational augmentations at training time and evaluating the resulting NN on an ensemble of coordinate systems at test time.

**Strengths:**

- The incorporation of rotational symmetries into NNs applied to point data seems very important to the application of DL to atomic data as it may reduce the data hunger of NNs and improve generalization.
- The  approach using an ensemble of coordinate systems seems innovative and makes sense.


**Weaknesses:**

- What is the typical number of coordinate systems in eq 3? Since each coordinate system is defined by three atoms, I guess it becomes very large very quickly.
- I wonder whether using y_s is even necessary. It seems to greatly increase the runtime at inference while not really adding anything in terms of performance. I.e. Simple training with rotational augmentation seems to yield all of the performance gains. Could the authors elaborate on this? I think it would be necessary to clearly demonstrate the usefulness of this addition
- Are competing methods also using rotational augmentations?
- -Do the authors have any idea why the SOTA in tab 1 is so much worse on MnO & HEA?


**Questions:**

see weaknesses

---

> ### Author Rebuttal · Authors · 2023-08-10
>
> **W1: Number of coordinate systems**
> The reviewer is correct that a naïve implementation of our core idea would involve summing over a very large number of coordinate systems. Our symmetrization scheme, however, doesn't suffer from this problem. The reason is that not all triplets of atoms are used to construct coordinate systems. Instead, we select only a few closest neighbors to the central atom. This is achieved by defining in a smooth way an adaptive cutoff radius that is as small as possible but sufficient to define at least one "good" (where the corresponding triplet of atoms is sufficiently non-collinear) coordinate system.
>
> The overall pipeline of our symmetrization protocol is quite complicated. Thus, for better readability, we decided to first explain (in Sec. 5, "Ensemble of coordinate systems") the simplest possible idea, which, as the reviewer notes, suffers from high computational cost and the presence of numerically unstable corner cases. Next, we provide several optimizations and solutions to all corner cases. The main ideas are sketched out in the paragraph "Adaptive cutoff and limiting cases" in sec. 5 of the main text. Details of the implementation are given in the Appendix F.4, and potential trade-offs between computational cost and smoothness are discussed in Appendix F.6. Empirically, for the parameters we used in our benchmarks our algorithm selects between 1 and 10 coordinate systems per environment.
>
> **W2: Need of symmetrization**
> We clarify the purpose of our symmetrization scheme in our global response. In short, the primary goal of our symmetrization protocol is not to further increase the accuracy. Instead, it is required for the very applicability of a model to practical simulations in atomistic modeling, a domain which considers exact compliance with physical symmetries as a basic requirement. The notion that models should be *intrinsically* equivariant is a constraint that has directed, and perhaps slowed down, the progress of ML applications to this domain. To sum up, given this domain requirement, without our symmetrization scheme, it is unlikely that PET-like models would be used in practical applications, despite their excellent accuracy. With our symmetrization scheme, PET becomes exactly invariant with respect to rotations and, thus, is applicable for atomistic modeling.
>
> In our manuscript, we described this in the introduction, but we acknowledge that our description might not be sufficiently clear, especially for readers from a different domain. We will do our best to provide a more compelling explanation in a revision of our manuscript.
>
> **W3: Rotational augmentation in competing methods**
> Competing methods are exactly, and intrinsically, rotationally invariant. Thus, applying rotational augmentations to them would change nothing at all. For instance, one of the methods [39] relies on distances between the atoms, angles, and dihedral angles. If an atomic configuration is rotated, the distances between the atoms and internal angles stay precisely the same, and so it is irrelevant whether rotational augmentation is used or not. To sum up, competing methods cannot be improved by using rotational augmentations (and in fact they are usually advertised for not requiring augmentation); thus, the comparison is fair.
>
> **W4: SOTA on MnO and HEA**
> The case of MnO corresponds to magnetic potentials, which is a more complicated case compared to the usual task in atomistic machine learning; see more details in [92]. Only few models currently support this setup. For instance, Dimenet, Gemnet, and MACE have not yet been reported to be extended to magnetic case and benchmarked on a magnetic dataset such as MnO. We acknowledge that if extended to a magnetic setup, these, or maybe some other, models might perform noticeably better than the current SOTA[92] on the MnO dataset. But this hasn't happened at the time of the submission, and moreover, nor until now, to the best of our knowledge. This aspect highlights the strength of PET, that it can be extended to nonstandard setups with very little effort. In addition, the current SOTA[92] on MnO is also a recent model (from 2021) that is used successfully in practical application, and outperforming it almost 3 times highlights the overall excellent performance of PET. To sum up, we regard the superior performance of PET on the MnO dataset as a strength, not a weakness.
>
> The HEA dataset has been very recently introduced, and is characterized by extreme levels of chemical diversity, with up to 25 chemical elements simultaneously considered in a single structure. The architecture proposed in the original paper relies heavily on physical priors to improve robustness and transferability. The Reviewer will notice how, in the discussion of the results, we emphasise how the dramatic increase in validation set performance is accompanied with a deterioration of accuracy in strongly extrapolative tasks. We believe this to be an interesting observation to guide future improvements of the PET architecture.
> Besides these examples, that serve as much as case studies for specific dataset features as much as empirical benchmarks, we also report a comparison with the COLL and HME21 datasets, to which Dimenet, Gemnet, and MACE have been also applied. The difference in performance between these models and PET is not as large (also reported in Table 1), but also in this case our model yields remarkable performance, particularly for the COLL dataset that was originally proposed by the authors of Dimenet and Gemnet.
>
> **References**
> Reference numbers match the ones in our manuscript.

---

> > ### Comment · Reviewer_oCKF · 2023-08-13
> >
> > I thank the authors for addressing my concerns sufficiently. I am leaning towards increasing my score. This will also depend on discussions with other reviewers.

---

> > > ### Author Response · Authors · 2023-08-14
> > >
> > > We thank the reviewer oCKF for reading our responses. We remain available for further clarifications, if needed.

---

### Official Review · Reviewer_QskA · 2023-07-03

**Soundness:** 3 good
**Presentation:** 3 good
**Contribution:** 3 good
**Rating:** 6
**Confidence:** 3

**Summary:**

This paper presents a novel posteriori rotational equivariant protocol that can be applied to any existing 3D models, irrespective of their initial equivariance to rotations. This protocol ensures not only rotational equivariance but also the preservation of other important model properties, such as permutation equivariance and expressiveness. Additionally, the paper introduces the Point Edge Transformer architecture, which achieves state-of-the-art performance on several benchmarks. The general equivariant design results in a modest performance improvement, highlighting the fact that exact equivariance is not always essential in certain learning tasks.

**Strengths:**

1. This paper alleviates the need to incorporate the rotational equivariance into the model but designs a general method on the top of the trained $3D$ model.  The method is proven to be strictly equivariant to 3D rotations, allowing for the utilization of different non-equivariant 3D models and enabling broad applications across various 3D tasks.

2. Through experimental analysis, this paper demonstrates that exact equivariance of the model is not always necessary for certain atomic applications. They show that by employing a powerful model and rotation augmentation techniques, non-intrinsically equivariant models outperform their counterparts. This finding provides valuable insights into the essential factors to consider when designing models for atomic applications.

3. This paper is well-written, and the experimental results effectively support the authors' argument.

**Weaknesses:**

1. The proposed equivariant protocol achieves equivariance but comes with increased computational requirements, particularly in scenarios with large sample sizes. Moreover, exact rotation equivariance may be affected by the presence of noise in the data, thus introducing potential sensitivity issues.

2. The proposed model may not applied to the 3D models like pointnet and pointnet++, which doesn't include the interaction of information between the points.

3. As mentioned in the paper, for GNN network, their method either requires more computation to go over the whole neighboring points, or  involves the redesign of the message-passing mechanism.





**Questions:**

I'm not familiar with the topic of atomic-scale applications. Therefore I will ask more questions about the equivariance part.

1. As stated in the limitation, I think the equivariant design might be sensitive the noise in the sample, please correct me if I'm wrong. Could the authors provide some experiments to validate its robustness?

2. Could the model be applied in large point cloud application such as shape segmentation or scene prediction in computer vision? I'm curious the scalabilty of the model and more applications of the model.

3. Since the authors show that in some tasks the equivariance is not the essential part. Could the authors compare their equivariant model with the other equivariant models in some tasks where the equivariance ideed is important, for example, missing atom prediction in tensor field networks?


**Limitations:**

The authors adequately addressed the limitations.

---

> ### Author Rebuttal · Authors · 2023-08-10
>
> **W1: Computational cost**
>  It is true that our method achieves equivariance at the cost of increased computational requirements. However, we made a significant effort to reduce this overhead. Because of the adaptive inner cutoff radius (described in appendix F.4.), the number of coordinate systems per each central atom doesn't depend on sample size, nor on the number of neighbors included in each local environment, and is typically between 1 and 10. A non-equivariant model is simpler, and can be usually implemented very efficiently, so the overall cost for an equivariant prediction can still be low.
>
> Our symmetrization scheme always achieves exact equivariance in a rigorous mathematical sense. Simultaneously it preserves the smoothness of the prediction. Thus, it is not affected by the noise in the data. See our response to related Q1, where we describe the requested numerical experiment to demonstrate this.
>
> **W2: Applicability to pointnet**
> Our symmetrization scheme can be applied to pointnet. As for pointnet++, we, indeed, are not aware of how to make this architecture as a whole applicable to atomistic modeling. However, the only obstacle pertains to downsampling layers, not to the ones that do interaction of information between the points. Moreover, due to the mostly local nature of atomistic interactions, we expect that one can remove downsampling layers from the pointnet++ architecture without a noticeable drop in accuracy. After such a modification, our symmetrization protocol can be applied to pointnet++. More details can be found in Appendix G.2.
>
> **W3: Message passing architectures**
> It is correct that the overhead when applying our symmetrization protocol to message passing architectures is steeper. As we discuss, it is possible to avoid this by re-designing the message-passing mechanism, and we are convinced that this can be done without substantially affecting the performance of the model. We provide a discussion of several approaches to do this in the manuscript in Appendix F.10. We think however that it would be impossible to perform a thorough investigation of these ideas in a manuscript that already discusses the symmetrization algorithm and benchmarks of a novel architecture.
>
> **Q1:** Our symmetrization scheme produces predictions that are smooth and exactly equivariant in a rigorous mathematical sense. Thus, it is robust to noise. We have conducted two numerical experiments to verify each of the statements.
>
> The first one shows that the symmetrized model is indeed exactly invariant. We have empirically measured the discrepancy between predictions for different rotations for single (torch.float32) and double (torch.float64) precisions. We find that for single precision evaluation, the rotational discrepancy is 1.1e-7, while for the torch.float64, the rotational discrepancy is 1.86e-15. The magnitude of the predictions themselves was designed to be of the order of 1e0. Such a dramatic difference, along with an overall minimal rotational discrepancy for torch.float64, strongly indicates that machine precision is the only source of difference in predictions for different rotations.
>
> The second experiment verifies the smoothness of our construction. We add random noise to the positions of all the points in an input point cloud and measure how much predictions change. Figure 1 in the figure response (the pdf file attached to the global response) illustrates that when the L2 norm of the noise converges to zero, so does the difference in predictions. Thus, this experiment verifies the smoothness of the symmetrized model.
>
> **Q2:** Yes, the model can be applied for point cloud applications beyond atomistic systems. The computational cost of the model scales linearly with the input size. Thus, it is, in principle, applicable to larger point clouds. We are also interested in the performance of the model for tasks in computer vision. Therefore this likely might be a direction of future research.
>
> **Q3.** Our experiments show that a non-equivariant model can achieve very good accuracy if trained with rotational augmentations, i.e. that *approximate* equivariance is enough to improve accuracy. For the suggested task of missing atom prediction, the target, i.e., the location of a missing atom, is not invariant. In other words, it is moving together with an input object. Thus, the model also must produce a non-invariant prediction that transforms in a certain manner with shifts and rotations of an input point cloud.
>
> Our benchmarks section already contains an experiment with similar settings, i.e. dipole moment prediction on the QM9 dataset. A dipole moment is a vector rigidly attached to a molecule. This vector rotates together with an input molecule. Thus, the model also must predict different outputs for different orientations. We achieve state-of-the-art on this benchmark, outperforming all the other models, as it is illustrated in Fig. 3 f. This is achieved by modifying a rotational augmentation strategy during the training. In addition to the random rotation of an input point cloud, the same rotation should be applied to a target property. We also note that our symmetrization scheme allows for ensuring exact vectorial equivariance during inference. This means that after applying our symmetrization protocol, the prediction of the overall model transforms exactly as a vector with rotations of an input point cloud. We believe that this numerical experiment is sufficient to demonstrate the performance of our model for a task where equivariance is crucial.

---

### Official Review · Reviewer_mhUe · 2023-07-05

**Soundness:** 3 good
**Presentation:** 2 fair
**Contribution:** 3 good
**Rating:** 5
**Confidence:** 3

**Summary:**

This paper presents a novel protocol for general symmetrization that adds rotational equivariance to any given 3D point-cloud processing model while preserving symmetries with respect to translations and permutations.
Applying the protocol to a model a-posteriori (i.e., at inference) makes the model rigorously rotation-equivariant.
To demonstrate the potential use of the protocol, the authors present Point Edge Transformer (PET).
Albeit not intrinsically equivariant, it achieves state-of-the-art performance on several benchmarks of molecules and solids.

---------------------------------------
POST REBUTTAL

Given the clarification the authors provided by the rebuttal, as well as the additional demonstration of the robustness of their proposed method wrt. noise, I have raised my score to 5.

**Strengths:**

S1. The proposed idea of training a model that is intrinsically non-SO(3)-equivariant and making it equivariant during inference is novel, and its mathematical presentation appears correct.

S2. The proposed experimental evaluation is extensive and substantial.

**Weaknesses:**

W1. Contributions: The main contribution the authors claim is the equivariance scheme.
Unfortunately, there is not much in the article to support this. Instead, the bulk of the improvement appears to come from the PET model, which is already sufficiently equivariant across multiple tasks.
It is unclear what benefits smoothness gave in practice (I might have missed something in the experiment part).

W2. Presentation: The presentation is a bit convoluted and seems too verbose; the provided Appendix contains many important details, e.g., about the PET architecture, that should instead be in the main paper.  Furthermore, I had a hard time finding a precise definition of the problem in the paper. The paper addresses an important question of rotational equi- and invariance, but its vague presentation makes it difficult to appreciate the contributions.

W3. Related work: Somewhat lacking related work on geometric deep learning. There is a detailed overview of related work on GNNs, but almost no discussion about group equivariant CNNs (G-CNNs). It is unclear how different the proposed model is and if the proposed protocol could be applied to such models.

**Questions:**

Q1. In the abstract, it says (line 11), "We propose a general symmetrization protocol that adds rotational equivariance to any given model". However, the symmetrization seems to assume a base model based on the atom-centric local coordinate system. How common is this? Does this assumption cover the related work, or are there methods that do not use this?

Q2. In the analysis (F.8), it is said that one can improve performance further by introducing extra rotations at inference (not on input but on extracted coordinate systems, so the model is still technically invariant/equivariant). I would like to know why this improves performance.
If I understood correctly from the training description, it is done in a non-equivariant way with rotational augmentation. Is this the reason for the improvement?

Q3. Given that the most common metric is MAE, have the authors tried using MAE as training loss?

Q4. Figure 2: $\bar{x}$ seems to refer to edges between atoms in $A_i$, but as far as I can see, $x$ has not been introduced before. The same applies to $E$ and $M$ and $H$. These terms should be clearly introduced in the text before the figure, not in the Appendix.

Q5. It appears that **r**$_i$ is the 3D position per atom, and $A_i$ is a group of atoms centered around **r**$_i$. However, it was unclear how features per atom are extracted from the beginning. Are different atoms treated as one-hot embeddings?

**Limitations:**

The authors adequately address the limitations and broader impact.

---

> ### Author Rebuttal · Authors · 2023-08-10
>
> **W1** The reviewer is correct in that the bulk of the accuracy improvement comes from the PET model. However, accuracy is not the primary reason to enforce symmetry or smoothness in our primary domain of application. We discuss this point in our global response, but in short the physical constraints in atomistic modeling make exact equivariance and smooth differentiability a strong requirement.
>
> Performance in benchmark examples is important, and an objective metric to compare models, but most practitioners in atomistic modeling would choose a rigorously equivariant model over one that is only approximately so, even if the latter was more accurate on benchmarks.
>
> **W2** The space constraints forced us to move to the Appendices the detailed description of our approaches, but we had tried to keep the core ideas in the main text. This might give the impression of a somewhat superficial treatment, and we will attempt to highlight more the key ideas. Similarly, it appears that we have not been able to convey the importance of exact equivariance in our (and related) research fields. Besides further clarifying this problem in the abstract and introduction, it might be useful to reiterate this motivation in the discussion and conclusions, to help readers from other communities who may be more focused on benchmark performance as the primary metric of success. In addition, we agree with the reviewer that the main text should be self-consistent in terms of notation, and we'll make sure to do that in the next revision
>
> **W3** Section 2 of our manuscript contains a review of group equivariant architectures relevant to the scope of our paper. Namely, it discusses such models as [41], [42], [43], [17], and [40].
> Reference R1 is less relevant, as it focuses on 2D symmetries whereas our focus is on the continuous 3D rotations. Still, we could certainly add a reference to this foundational work.
>
> The PET model does not belong to this class of models, and therefore requires application of our symmetrization protocol to become exactly equivariant. There would be no need to apply symmetrization to an architecture that is intrinsically equivariant, and if one did so there would be absolutely no change in the predictions. However, our experiments on PET show that, contrary to the common wisdom in this application domain, models with built-in equivariance are not necessarily more accurate than models that are not. Further performance improvements might be possible in this extended design space: thanks to our exact symmetrization protocol, these improved models would be readily applicable to atomistic simulations.
>
> **Q1** Our symmetrization scheme needs some reference point in order to add rotational equivariance on top of any base model. In principle, this reference point might be arbitrary, for instance, the center of mass of the whole system. The use of all atomic positions as multiple reference points is not a restriction, but a choice that is associated with the nature of target properties in atomistic machine learning. We focus on this case because very often, and in particular for all the models we describe in section 2, predictions are given by an atom-centered decomposition of the target, but it is not a hard requirement.
>
> **Q2** Yes, the training is done in a non-equivariant way with rotational augmentations. We would say that the reason for the improvement is that the base model is not rotationally equivariant. For a non-equivariant model, one can apply a rotational augmentation strategy also during testing. This can be done by generating N random rotations, running the model for each of them, and next averaging the predictions. The larger the N, the better the accuracy. The expectation of the error monotonically decreases with N to some limit but never reaches it for any finite N.
> The functional form of our symmetrization protocol is a weighted average of the model's predictions for different coordinate systems. Thus, it can also be seen as some form of rotational augmentations strategy during testing. Crucially, it uses a finite number of reference systems, that are chosen so that it achieves exact equivariance with a finite (and small) number of coordinate systems. However, it might not reach the asymptotic accuracy discussed above: in this case, if one artificially increases the number of coordinate systems, as described in F.8, the error should (slighlty) decrease.
>
> **Q3** No, we have always used a weighted sum of MSE losses in energies and forces. However, we used a checkpoint with the best MAE on a validation dataset for a final prediction, if MAE metrics was used.
>
> **Q4** Yes, $\tilde{x}_{ij}^{k}$ is a hidden representation, also a message, associated with an edge between atoms i and j after the layer k. We thank the reviewer for the possibility to improve our manuscript by pointing us to this lack of definition in the main text. In revising our manuscript, this definition will be moved to the main text from the Appendix.
>
> **Q5** Features associated with each neighbor are given by encoding:
> 1) A 3D displacement vector from the central atom to the neighbor
> 2) Embedding of the atomic species of the neighbor atom
> 3) For each layer but the first, an input message
>
> More details can be found in Appendix A. We use optimizable embedding, where an abstract vector of a dimensionality $d_{PET}$ is assigned to each atomic specie. $d_{PET}$, which controls the dimensionality of virtually all hidden representations of our model, was set to 128 for most of our computational experiments and to 256 for a couple of them. The number of parameters in a single embedding layer is $d_{PET}$ multiplied by the number of unique atomic species in a dataset. Such an embedding is exactly equivalent to a linear layer applied on top of one-hot embedding.
>
> **References:**
>
> [R1] Cohen, T., & Welling, M. "Group equivariant convolutional networks."
>
> All the other references coincide with the ones in the main text.

---

> > ### Comment · Reviewer_mhUe · 2023-08-12
> >
> > I thank the authors for their detailed rebuttal!
> >
> > If I understand it correctly, the main argument the authors make is that equivariance per se is important in many application areas, especially atomistic modeling.
> > In this case, the experiments should have a different focus at the beginning: namely, demonstrating how exact the equivariance attained with the proposed protocol is.
> >
> > Given the clarification the authors provided by the rebuttal, as well as the additional demonstration of the robustness of their proposed method wrt. noise, I'm willing to raise my score to 5.

---

> > > ### Author Response · Authors · 2023-08-14
> > > **Exact equivariance and choice of benchmarks**
> > >
> > > We thank the reviewer for reading our response and for the further comments!
> > >
> > > **Exact equivariance**
> > > The reviewer is correct; equivariance *per se* is considered essential and is almost universally enforced in the field. We did not include numerical tests of equivariance because the algorithm is designed to make predictions *exactly* equivariant, up to machine precision. However, following our discussion with the Reviewers, we agree it would be worthwhile to add a numerical demonstration similar to the one we did in our responses, as a validation that our implementation is correct. We will do this upon revision.
> > >
> > > **Choice of benchmarks**
> > > The reason we dedicated our experiment section to assessing the performance of our proposed model, PET, on various datasets pertains to a common (mis)belief in the community of atomistic machine learning that exact intrinsic equivariance is not only required for physical consistency but is also necessary to achieve good accuracy. Thus, without such an exhaustive demonstration of PET's excellent and often superior performance, many researchers from this community would doubt the usefulness of our symmetrization method, arguing that there is no reason to fit and next symmetrize non-equivariant models, given that intrinsically equivariant frameworks are already available. With such a demonstration, we have provided strong indications of the opposite, that it is possible to construct better models within an enlarged design space without the burden of incorporating rotational equivariance intrinsically.

---

### Official Review · Reviewer_r8ta · 2023-07-07

**Soundness:** 2 fair
**Presentation:** 2 fair
**Contribution:** 2 fair
**Rating:** 5
**Confidence:** 2

**Summary:**

The paper suggests a network design tackling chemical and materials modeling. The paper identifies rotational symmetries as lacking in existing solutions. The suggested approach is based on symmetrization, i.e., projecting a general function to be equivariant by averaging it over aligning rotations. In addition, a transformer-based architecture is proposed.

**Strengths:**

The idea to enforce smoothness with other symmetries such as rotational symmetries seems to be valuable, and indeed missing from existing solutions.

**Weaknesses:**

The main weaknesses of the paper are related to its level of readability, the quality of its exposition, and its mathematical formulation. In what follows, I will share some relevant examples.

It is understood that the selected approach to incorporate rotation equivariance is symmetrization. However, the text does not make it clear what are the challenges in doing so and why the starting point chosen is using frames from pairs. If that is the starting point, it is not entirely clear what is the next point. In addition, it is not shown why this symmetrization is indeed equivariant. In addition, there so no clear and simple mathematical formulation of how the aligning rotations for the symmetrization are selected. Furthermore, all the input and output dimensions are missing from all the functions stated in the paper. Figure 2 is hard to decipher.

It is hard to follow the experiment section. I would expect to see some ablation regarding the symmetrization technique. I would expect to see the symmetrization tested with other architectures than the transformer as well. Furthermore, it seems that symmetrization has a minor effect on the model’s generalization capacity, making the benefit from it, given its computational budget increase, questionable.

Lastly, the contribution of the proposed transformer architecture seems to be orthogonal to the challenge the paper set to tackle.

**Questions:**

I would appreciate any response from the authors regarding the weakness stated above.

**Limitations:**

Yes

---

> ### Author Rebuttal · Authors · 2023-08-10
>
> A large part of the perceived weaknesses is, in our opinion, connected to the fact that were not sufficiently clear in explaining that the issue of achieving exact equivariance is mainly related to the very applicability of the models for atomistic simulations. A slight improvement in accuracy is just a fortunate side effect. We discuss this point in the global response, but we would like to address specifically some of the points raised here.
>
>  > However, the text does not make it clear what are the challenges in doing so and why the starting point chosen is using frames from pairs.
>
> The challenge is to achieve exact rotational equivariance while preserving permutation equivariance and smoothness of the model, as we discuss in the introduction and in Sections 4 and 5. Defining a reference system that is rigidly attached to a group of atoms is a straightforward way to obtain equivariant predictions. It is however very hard to do so while preserving smoothness. We review two approaches that define a single coordinate system to ensure rotational equivariance. The first uses a single pair of two closest neighbors, while the second utilizes eigenvectors of the tensor of inertia. Both methods lead to discontinuous predictions. Thus, our method, which sums over an ensemble of local coordinate systems, is the first one which does rotational symmetrization while preserving smoothness. This idea is the “starting point” of our construction.
>
> > If that is the starting point, it is not entirely clear what is the next point.
>
> The next point is a discussion of the modifications that have to be applied to make this idea computationally efficient, by reducing the number of coordinate systems without affecting differentiability. They are described in the paragraphs "Coordinate weighting" and "Adaptive cutoff and limiting cases." Details of implementation are given in Appendix F.
>
> > In addition, it is not shown why this symmetrization is indeed equivariant.
>
> The prediction is computed as a weighted average of predictions of the base model for several coordinate systems rigidly attached to an input point cloud. It means that all the used coordinate systems rotate together with the given input. This, in turn, leads to equivariance. Mathematically, one can consider that in Eq. (3) every rotation applied to the structure would automatically be applied to all the $\hat{R}_{jj'}$, which directly translates into proof of equivariance.
>
> > In addition, there so no clear and simple mathematical formulation of how the aligning rotations for the symmetrization are selected.
>
> Each coordinate system used for symmetrization is built based on a triplet of atoms. One of the atoms is always the central one, and two others are neighbors within an adaptive cutoff radius. This is stated in Section 5, “$\hat{R}_{jj'}[\cdot]$ indicates the operator that rotates its argument in the coordinate system defined by the neighbors $j$ and $j'$ within the $i$-centered environment $A_i$”.
> The gory details of how we reduce the number of coordinate systems in a smooth manner is discussed in detail in the Appendix F.
>
> > Furthermore, all the input and output dimensions are missing from all the functions stated in the paper.
>
> We apply the common convention that vectorial quantities are typeset in boldface. All other quantities and functions in the main text are scalars. The only exception is the dimensionality of the output of the base model in the description of our symmetrization scheme in section 5. However, since our symmetrization method is very general, this dimensionality can be arbitrary. In the Appendices, we indicate the dimensionalities of all multidimensional objects.
>
> > It is hard to follow the experiment section. I would expect to see some ablation regarding the symmetrization technique. Furthermore, it seems that symmetrization has a minor effect on the model's generalization capacity, making the benefit from it, given its computational budget increase, questionable.
>
> We perform this ablation study, as we report both “raw” and symmetrized accuracy for each benchmark. The purpose of our symmetrization technique is not to improve accuracy. Instead, it is necessary for the very applicability of ML models to atomistic applications. A minor improvement in the model's accuracy is just a fortunate side effect. We provide more details in our global response.
>
>  > Lastly, the contribution of the proposed transformer architecture seems to be orthogonal to the challenge the paper set to tackle.
>
> As stated at the beginning of Section 6, we introduce PET to “demonstrate that lifting the design constraint of rotational equivariance makes it possible to construct better models.” The requirement to have exact equivariance has led to almost exclusive focus on models that are intrinsically equivariant, which greatly restricts the design space for new architectures. Being able to achieve exact equivariance *a posteriori* lifts these design constraints, and we need PET to prove that there is a practical advantage in doing so.

---

> > ### Comment · Reviewer_r8ta · 2023-08-21
> > **rebuttal**
> >
> > I thank the authors for a detailed rebuttal. Some of my concerns were addressed, specifically, the one clarifying that the application in hand requires equivariant/consistency in predictions. I will raise my score to 5. Thank you.

---

> > > ### Author Response · Authors · 2023-08-21
> > >
> > > We thank the reviewer for reading our response!

---

### Author Rebuttal · Authors · 2023-08-10

We thank all reviewers for providing their comments, which highlight opportunities to improve the presentation of our work. We would like to start by addressing a misunderstanding that appears to recur in several places, concerning the relation between the two main contributions in our manuscript - the *a posteriori* symmetrization and the PET model.

**Exact equivariance is de facto a requirement for atomistic modeling**
As we should probably have made clearer, and as we will certainly clarify in later revisions, the main reason why several application domains, such as atomic-scale simulations, require exact equivariance is to preserve fundamental physical constraints, lack of which can lead to misleading end results (e.g. spontaneous ordering of molecules in an isotropic fluid). For this reason, equivariance that is exact in a rigorous mathematical sense is highly sought after. An approximate one, achievable by supplementing any model with rotational augmentations, would be considered insufficient by most practitioners. This is the primary reason why, at variance with other fields, the vast majority of ML models for atomistic modeling are built around intrinsically equivariant architectures.

**Non-equivariant models can outperform intrinsically equivariant ones**
One could argue that, given that such equivariant architectures exist, our symmetrization protocol solves a non-existent problem. This is why we devote a substantial part of our manuscript to describe and benchmark a non-equivariant model on atomistic-simulation datasets. The excellent performance of PET demonstrates that it can be beneficial to relax the requirement of intrinsic equivariance when designing ML models for atomistic simulation tasks, an observation that we suspect may come as surprising to many domain experts.

**There is a wealth of models and ideas that can now be applied to new domains**
There are dozens of algorithms for generic point clouds (some of which we summarize in Sec. 3, e.g. PointNet, PointNet++, O-cnn, Octnet, kd-network, Kpconv, PointMLP, PAConv, Point Transformer, PVT) that have never been applied to atomistic simulations because of lack of equivariance: this untapped potential for models, or architectural features, that can now be seamlessly applied to a new domain, is in our opinion a major outcome of our study.
The small performance improvement that comes from symmetrizing *a posteriori* the PET model is a fortunate side effect of a procedure whose primary goal is to make PET, and other models, applicable to domains of science that require exact compliance with fundamental physical requirements.

**Choice of benchmarks, and insights on architectural features**
Our model, PET, achieves state of the art performance on several benchmarks. For instance, it outperforms such models as DimeNet++ and GemNet on the COLL dataset, developed by the authors of the mentioned models. Another general consideration, however, and we would like to better highlight it in the manuscript, is that we picked most of our benchmarks not only as a way to demonstrate a favourable comparison with the state of the art, but also as a way to highlight some of the strengths and/or shortcomings of our model. For instance, MnO and the dipole models allow us to highlight the simplicity by which non-standard setups (namely atomic attributes associated with magnetism, and the vectorial nature of the target) can be incorporated in the model. The water dataset is well-suited for ablation studies on the receptive field of the model. The HEA dataset allows us to highlight the need to improve the extrapolative capabilities of PET. We believe that this choice adds an interesting dimension to our analysis, while some reviewers seem to have focused their comments primarily on the bare benchmark numbers.

**Summary of contributions**
To summarize, the impact of our work is best understood in the context of applications to chemical and materials simulations, for which exact equivariance and smoothness are considered a hard requirement, that have guided and somewhat restricted the development of geometric machine-learning schemes to frameworks that are intrinsically equivariant. In this context our manuscript contributes:
* A practical algorithm to make a generic point-cloud model exactly, rigorously equivariant: this makes multiple existing algorithms applicable to a new domain.
* A demonstration that relaxing the architectural constraint of equivariance can be beneficial in terms of model accuracy. This will come as a surprise to many practitioners, as “built-in” equivariance is usually expected to be also beneficial in terms of accuracy
* A novel architecture that is comparable or better than several state-of-the-art approaches, and that will likely find its own practical applications

---

### Decision · Program_Chairs · 2023-09-21

**Decision:**

Accept (poster)

**Comment:**

After the rebuttal discussion, all reviewers agreed to accept this work (one did so in a comment but did not update their review).  All recognize the importance of the problem tackled in the paper, although the amount of space in the paper for equivariance relative to PETS rightfully seems imbalanced given the focus of the work.

For the camera-ready version of the paper, please add the assorted changes and experiments introduced during the rebuttal phase.

(Also, the AC wonders why the "Vector Neurons" architecture and follow-on work isn't referenced here?)